# Closing the gap between SVRG and TD-SVRG with Gradient Splitting

**Arsenii Mustafin**  *aam@bu.edu*
*Department of Computer Science*
*Boston University*

**Alex Olshevsky**  *alexols@bu.edu*
*Department of Electrical and Computer Engineering*
*Boston University*

**Ioannis Ch. Paschalidis**  *yannisp@bu.edu*
*Department of Electrical and Computer Engineering*
*Boston University*

**Reviewed on OpenReview:** *https://openreview.net/forum?id=dixU4fozPQ*

## Abstract

Temporal difference (TD) learning is a policy evaluation in reinforcement learning whose performance can be enhanced by variance reduction methods. Recently, multiple works have sought to fuse TD learning with Stochastic Variance Reduced Gradient (SVRG) method to achieve a geometric rate of convergence. However, the resulting convergence rate is significantly weaker than what is achieved by SVRG in the setting of convex optimization. In this work we utilize a recent interpretation of TD-learning as the splitting of the gradient of an appropriately chosen function, thus simplifying the algorithm and fusing TD with SVRG. Our main result is a geometric convergence bound with predetermined learning rate of 1/8, which is identical to the convergence bound available for SVRG in the convex setting. Our theoretical findings are supported by a set of experiments.

## 1 Introduction

Reinforcement learning (RL) is a framework for solving sequential decision making environments. Policy evaluation is one of those problems, which seeks to determine the expected return an agent achieves if it chooses actions according to a specific stationary policy. Temporal Difference (TD) learning Sutton (1988) is a popular algorithm with a particularly simple form which can be performed in an online setting. TD learning uses the Bellman equation to bootstrap the estimation process and update the value function from each incoming sample or mini-batch. As all RL methods, tabular TD learning suffers from the "curse of dimensionality" when the number of states is large, motivating parametric approximations of the value function.

Despite its simple formulation, theoretical analysis of approximate TD learning is subtle. There are a few important milestones in this process, one of which is the work in Tsitsiklis & Van Roy (1997), where asymptotic convergence guarantees were established. More recent advances include Bhandari et al. (2018), Srikant & Ying (2019) and Liu & Olshevsky (2021). In particular, Liu & Olshevsky (2021) shows that TD learning might be viewed as an example of gradient splitting, a process analogous to gradient descent.

TD-leaning has an inherent variance problem: the variance of the update does not go to zero as the method converges. This problem is also present in a class of convex optimization problems where the objective function is a sum of functions and *Stochastic Gradient Descent (SGD)*-type methods are applied Robbins & Monro (1951). Such methods proceed incrementally by sampling a single function, or a mini-batch

of functions, to use for stochastic gradient evaluations. *Variance reduction techniques* were developed to address this problem and yield faster convergence, including Stochastic Average Gradient (SAG) (Schmidt et al., 2013), SVRG (Johnson & Zhang, 2013) and SAGA (Defazio et al., 2014). Their distinguishing feature is that they converge geometrically.

Previous research has analysed the application of variance reduction technique to TD updates in two problem settings: (*i*) a pre-sampled trajectory of the *Markov Decision Process (MDP)* (finite sample), and (*ii*) when states are sampled directly from the MDP (online sampling). We briefly mention the most relevant works in both veins. In the online sampling setting, the first attempt to adapt variance reduction to TD learning was made in Korda & La (2015). Their results were discussed by Dalal et al. (2018) and Narayanan & Szepesvári (2017); Xu et al. (2020) provided further analysis of such approaches and showed geometric convergence for the so-called *Variance Reduction Temporal Difference learning (VRTD)* algorithm for both Markovian and i.i.d. sampling; Ma et al. (2020) applies the variance reduction technique to Temporal Difference with Correction. However, both Xu et al. (2020) and Ma et al. (2020) achieve total complexity better than $1/\epsilon$ for the policy evaluation problem, which is not possible in the setting of this paper (see Appendix A for details).

The finite sample setting was analysed in Du et al. (2017), where authors directly applied SVRG and SAGA to a version of policy evaluation by transforming it into an equivalent convex-concave saddle-point problem. Since their algorithm uses two sets of parameters, in this paper we call it Primal-Dual SVRG or PD-SVRG. Their results were improved in Peng et al. (2020) by introducing inexact mean path update calculation (Batched SVRG algorithm).

## 1.1 Motivation and Contribution

The previous analysis of finite sample settings and both cases of online sampling has demonstrated the geometric convergence of the algorithm. However, this convergence has been established separately with different proof strategies, and several unsatisfying aspects persist. A prominent concern is the high complexity in all scenarios: convergence times derived for variance reduction in temporal difference learning not only show a quadratic relationship with the condition number (ratio of largest to smallest eigenvalues of a matrix) but also include additional factors related to the condition number of certain diagonalizing matrices. Such complexities, especially in the context of ill-conditioned matrices typical in reinforcement learning, lead to prohibitive sample complexities even for straightforward problems. For instance, in a simple Markov Decision Process (MDP) with 400 states and 10 actions, the batch size required to ensure convergence is impractically large using the bounds from previous work, as illustrated in Table 2 (second and third rows of the table) and further discussed in Appendix J.1.

Additionally, there is a qualitative discrepancy in the current results: the current analysis of the SVRG-enhanced TD algorithm requires complexity that is *quadratic* in terms of the condition number, which does not align with the complexity of classical SVRG in the convex setting with its *linear* dependency on the condition number. This gap, not addressed by the previous literature, remains an unresolved question.

In this paper we analyze the convergence of SVRG applied to TD (for convenience we call it TD-SVRG) in both finite sample and online sampling cases. Our theoretical results are summarized in Table 1. Our key contributions are:

- For the finite sample case, we show that TD-SVRG has the same convergence rate as SVRG in the convex optimization setting. In particular, we replace the quadratic scaling with the condition number by linear scaling and remove extraneous factors depending on the diagonalizing matrix. Notably, we use a simple, pre-determined learning rate of 1/8 to do this.

- For i.i.d. online sampling, we similarly achieve better rates with simpler analysis. Again, our analysis is the first to show that TD-SVRG has the same convergence rate as SVRG in the convex optimization setting with a predetermined learning rate of 1/8, and a linear rather than quadratic scaling with the condition number. Similar improvement is obtained for Markovian sampling.

- Our theoretical findings have significant practical implications: Previous analyses for both finite sample and online sampling scenarios require batch sizes so large as to be impractical. In contrast,

Table 1: Comparison of algorithmic complexities, where $\epsilon$ is a desired expected accuracy, $\lambda_A$ is a minimum eigenvalue of the matrix $A$, $\pi_{\min}$ is a minimum state probability of the MDP stationary distribution, $\gamma$ is a discount factor and $N$ is a dataset size. The complexity is reported as the number of samples required to shrink a distance function on average by a factor of $\epsilon$, where the distance function is a quadratic of the quantity $\theta - \theta^*$, similar to previous works (Du et al., 2017; Xu et al., 2020). The definitions of other quantities and a table with additional algorithms and details of the comparison might be found in Appendix K

| Type | Algorithm | Complexity | |
|------|-----------|------------|--|
| | | Feature case | Tabular case |
| Finite | PD-SVRG | $\mathcal{O}\left(\left(N + \frac{\kappa^2(C)L_G^2}{\lambda_{\min}(A^T C^{-1} A)^2}\right)\log(\frac{1}{\epsilon})\right)$ | $\mathcal{O}\left(\left(N + \frac{1}{(1-\gamma)^2 \pi_{\min}^4}\right)\log(\frac{1}{\epsilon})\right)$ |
| Finite | Our | $\mathcal{O}\left(\left(N + \frac{1}{\lambda_A}\right)\log(\frac{1}{\epsilon})\right)$ | $\mathcal{O}\left(\left(N + \frac{1}{(1-\gamma)\pi_{\min}}\right)\log(\frac{1}{\epsilon})\right)$ |
| i.i.d. | TD | $\mathcal{O}\left(\frac{1}{\lambda_A^2 \epsilon}\log(\frac{1}{\epsilon})\right)$ | $\mathcal{O}\left(\frac{1}{(1-\gamma)^2 \pi_{\min}^2 \epsilon}\log(\frac{1}{\epsilon})\right)$ |
| i.i.d. | Our | $\mathcal{O}\left(\frac{1}{\lambda_A \epsilon}\log(\frac{1}{\epsilon})\right)$ | $\mathcal{O}\left(\frac{1}{(1-\gamma)\pi_{\min}\epsilon}\log(\frac{1}{\epsilon})\right)$ |
| Markovian | VRTD | $\mathcal{O}\left(\frac{1}{\epsilon \lambda_A^2}\log(\frac{1}{\epsilon})\right)$ | $\mathcal{O}\left(\frac{1}{(1-\gamma)^2 \pi_{\min}^2 \epsilon}\log(\frac{1}{\epsilon})\right)$ |
| Markovian | Our | $\mathcal{O}\left(\frac{1}{\epsilon \lambda_A}\log^2(\frac{1}{\epsilon})\right)$ | $\mathcal{O}\left(\frac{1}{(1-\gamma)\pi_{\min}\epsilon}\log^2(\frac{1}{\epsilon})\right)$ |

Table 2: This table gives the output of formulas from Table 1 on the simplest possible MDP (a random MDP) to show the magnitude of the improvement. Specifically, we compare theoretically suggested batch sizes for a random MDP with 400 states, 10 actions and $\gamma = 0.95$. Values in the first row indicate the dimensionality of the feature vectors. Values in the other rows show the batch size required by the corresponding method. Values are averaged over 10 generated datasets and environments.

| Method/Features | 6 | 11 | 21 | 41 |
|-----------------|---|----|----|----|
| TD-SVRG (ours) | 3176 | 6942 | 18100 | 54688 |
| PD-SVRG | $1.72 \cdot 10^{16}$ | $3.83 \cdot 10^{18}$ | $3.06 \cdot 10^{21}$ | $5.77 \cdot 10^{24}$ |
| VRTD | $5.41 \cdot 10^6$ | $2.53 \cdot 10^7$ | $1.63 \cdot 10^8$ | $1.58 \cdot 10^9$ |

our analysis leads to batch-sizes that are implementable in practice. A simple example of random MDPs illustrating this is given in Table 2.

- We conducted experimental studies demonstrating that our theoretically derived batch size and learning rate achieve geometric convergence and outperform other algorithms that rely on parameters selected via grid search, as detailed in Section 6 and Appendix J. Specifically, we have re-done earlier experiments from Du et al. (2017) and found that our TD-SVRG method with parameters coming from our theoretical analysis converges much faster than previous best SVRG based algorithm (PD SVRG): on average, it requires 132 times fewer iterations to contract by a factor of 0.5. Note that this comparison favors previous work due; specifically we use parameters from our theorems whereas previous work uses parameters selected by grid search. When we also run our algorithm with parameters chosen via grid search this disparity increases to 180 times.

To summarize, in every setting our key contribution is the reduce the scaling with a condition number from quadratic to linear, as well as to remove extraneous factors that do not appear in the analysis of SVRG in the convex setting. As described below, the final result matches the bounds that are known for the SVRG in the separable convex optimization setting. These theoretical results also lead to large gains in convergence speed.

## 2 Problem formulation

We consider a discounted reward Markov Decision Process (MDP) defined by the tuple $(\mathcal{S}, \mathcal{A}, \mathcal{P}, r, \gamma)$, where $\mathcal{S}$ is the state space, $\mathcal{A}$ the action space, $\mathcal{P} = \mathcal{P}(s'|s, a)_{s,s'\in\mathcal{S}, a\in\mathcal{A}}$ the transition probabilities, $r = r(s, s')$ the reward function, and $\gamma \in [0, 1)$ is a discount rate. The agent follows a policy $\pi : \mathcal{S} \to \Delta_{\mathcal{A}}$ – a mapping from states to the probability simplex over actions. A policy $\pi$ induces a joint probability distribution $\pi(s, a)$, defined as the probability of choosing action $a$ while being in state $s$. Given that the policy is fixed and we are interested only in policy evaluation, for the remainder of the paper we will consider the transition probability matrix $P$, such that: $P(s, s') = \sum_a \pi(s, a)\mathcal{P}(s'|s, a)$. We assume, that the Markov process produced by the transition probability matrix is irreducible and aperiodic with stationary distribution $\mu_\pi$.

The policy evaluation problem is to compute $V^\pi$, defined as: $V^\pi(s) := \mathbb{E}\left[\sum_{t=0}^\infty \gamma^t r_{t+1}\right]$, which is the expected sum of discounted rewards, where the expectation is taken with respect to the sampled trajectory of states. Here $r_t$ is the reward at time $t$ and $V^\pi$ is the value function, formally defined to be the unique vector which satisfies the Bellman equation $T^\pi V^\pi = V^\pi$, where $T^\pi$ is the Bellman operator, defined as: $T^\pi V^\pi(s) = \sum_{s'} P(s, s')\left(r(s, s') + \gamma V^\pi(s')\right)$. The TD(0) method is defined as follows: one iteration performs a fixed point update on a randomly sampled pair of states $s, s'$ with learning rate $\alpha$: $V(s) \leftarrow V(s) + \alpha(r(s, s') + \gamma V(s') - V(s))$. When the state space size $|\mathcal{S}|$ is large, tabular methods which update the value function for every state become impractical. For this reason, a linear approximation of the value function is often used. Each state is represented by a feature vector $\phi(s) \in \mathbb{R}^d$ and the state value $V^\pi(s)$ is approximated by $V^\pi(s) \approx \phi(s)^T\theta$, where $\theta$ is a tunable parameter vector. A single TD update on a randomly sampled transition $s, s'$ becomes:

$$\theta \leftarrow \theta + \alpha g_{s,s'}(\theta) = \theta + \alpha((r(s, s') + \gamma\phi(s')^T\theta - \phi(s)^T\theta)\phi(s)),$$

where the second equation should be viewed as a definition of $g_{s,s'}(\theta)$.

Our goal is to find a parameter vector $\theta^*$ such that the average update vector is zero

$$\mathbb{E}_{s,s'}[g_{s,s'}(\theta^*)] = 0,$$

where the expectation is taken with respect to sampled pair of states $s, s'$. This expectation is also called mean-path update $\bar{g}(\theta)$ and can be written as:

$$\begin{aligned}
\bar{g}(\theta) = \mathbb{E}_{s,s'}[g_{s,s'}(\theta)] &= \mathbb{E}_{s,s'}[(\gamma\phi(s')^T\theta - \phi(s)^T\theta)\phi(s)] + \mathbb{E}_{s,s'}[r(s, s')\phi(s)] \\
&:= -A\theta + b,
\end{aligned} \tag{1}$$

where the last line should be taken as the definition of the matrix $A$ and vector $b$. Finally, the minimum eigenvalue of the matrix $(A + A^T)/2$ plays an important role in our analysis and will be denoted as $\lambda_A$.

There are a few possible settings of the problem: the samples $s, s'$ might be drawn from the MDP on-line (Markovian sampling) or independently (*i.i.d.* sampling): the first state $s$ is drawn from $\mu_\pi$, then $s'$ is drawn as the next state under the policy $\pi$. Another possible setting for analysis is the "finite sample set": first, a trajectory of length $N$ is drawn from an MDP following Markovian sampling and forms dataset $\mathcal{D} = \{(s_t, a_t, r_t, s_{t+1})\}_{t=1}^N$. Then TD(0) proceeds by drawing samples from this dataset. Note that the definition of the expectation $\mathbb{E}_{s,s'}$ and, consequently, of matrix $A$ will be slightly different in these two settings: in the on-line sampling case probability of a pair of states $s, s'$ is determined by the stationary distribution $\mu_\pi$, and the transition matrix $P$; we define

$$A_e = \sum_{s\in\mathcal{S}}\sum_{s'\in\mathcal{S}} \mu_\pi(s)P(s, s')\phi(s)(\phi(s)^T - \gamma\phi(s')^T).$$

In the "finite sample" case, the probability of $s, s'$ refers to the probability of getting a pair of states from one particular data point $t$: $s = s_t, s' = s_{t+1}$, and the matrix $A$ is defined as:

$$A_d = \frac{1}{N}\sum_{t=1}^N \phi(s_t)(\phi(s_t)^T - \gamma\phi(s_{t+1})^T).$$

Likewise, the definition of $\bar{g}(\theta)$ differs between the MDP and dataset settings, since that definition involves $\mathbb{E}_{s,s'}$ which, as discussed above, means slightly different things in both settings.

In the sequel, we will occasionally refer to the matrix $A$. Whenever we make such a statement, we are in fact making two statements: one for the dataset case when $A$ should be taken to be $A_d$, and one in the on-line case when $A$ should be taken to be $A_e$.

We make the following standard assumptions:

**Assumption 2.1. (Problem solvability)** The matrix $A$ is non-singular.

**Assumption 2.2. (Bounded features)** $||\phi(s)||_2 \leq 1$ for all $s \in \mathcal{S}$.

These assumptions are widely accepted and have been utilized in previous research within the field Bhandari et al. (2018), Du et al. (2017), Korda & La (2015), Liu & Olshevsky (2021), Xu et al. (2020). Assumption 2.1 ensures that $A^{-1}b$ exists and the problem is solvable. At the risk of being repetitive, we note that this is really two assumptions, one that $A_e$ is non-singular in the on-line case, and one that $A_d$ is non-singular in the dataset case, which are stated together. Assumption 2.2 is made for simplicity and it can be satisfied by feature vector rescaling.

In our analysis we often use the function $f(\theta)$, defined as:

$$f(\theta) = (\theta - \theta^*)^T A (\theta - \theta^*). \tag{2}$$

We will use $f_d$ and $f_e$ notation for the dataset ($A = A_d$) and environment ($A = A_e$) cases respectively.

## 3 The TD-SVRG algorithm

Let us consider an optimization problem where the target function $f(\theta)$ is the sum of convex functions $f(\theta) = (1/N) \sum_{i=1}^{N} f_i(\theta)$, and the total number of functions $N$ is very large. This makes computing the full gradient $(1/N) \sum_{i=1}^{N} \nabla f_i(\theta)$ too costly for every update. Instead, during iteration $t$, we want to apply an update $g_t$ that is inexpensive to compute:

$$\theta_t = \theta_{t-1} + \alpha g_t,$$

where $\alpha$ is the learning rate. If we use $g_t = \nabla f_i(\theta)$ with a randomly chosen $i$, we obtain a standard SGD (stochastic gradient descent) algorithm. The challenge with SGD lies in its high variance. One common approach to mitigate this is by applying so-called variance reduction techniques: we instead update

$$\theta_t = \theta_{t-1} + \alpha v_t,$$

where

$$v_t = g_t - g_t' + \mathbb{E}[g_t'],$$

for some appropriately defined $g_t'$. The key idea here is that regardless of how we choose $g_t'$, we will have

$$E[v_t] = E[g_t],$$

so in expectation the update is the same. On the other hand, if we can choose $g_t'$ to be highly correlated with $g_t$, then the variance of $v_t$ will be substantially smaller than the variance of $g_t$.

There are several algorithms based on this idea, the most prominent of which are SAG (Schmidt et al., 2013), SAGA (Defazio et al., 2014), and SVRG (Johnson & Zhang, 2013). These algorithms propose different methods of constructing $g_t'$. In this work, we take our inspiration from the SVRG algorithm which suggests to choose $g_t$' to be the gradient of the function $f_i$ chosen at time step $t$, but estimated on a *previous* parameter vector $\tilde{\theta}$: $g_t' = \nabla f_i(\tilde{\theta})$, $\tilde{\theta} = \theta_{t'}$, $t' < t$.

The major drawback of this idea is that $\mathbb{E}[g_t'] = (1/N) \sum_{i=1}^{N} \nabla f_i(\tilde{\theta})$ – while being a full update for the parameter vector – is costly to compute, because the motivating scenario here involved large $N$. However,

---

**Algorithm 1** TD-SVRG for the finite sample case

---

    **Parameters** update batch size $M$ and learning rate $\alpha$.
    **Initialize** $\tilde{\theta}_0$.
    **for** $m' = 1, 2, ..., m$ **do**
        $\tilde{\theta} = \tilde{\theta}_{m'-1}$,
        $\bar{g}_{m'}(\tilde{\theta}) = \frac{1}{N} \sum_{s,s' \in \mathcal{D}} g_{s,s'}(\tilde{\theta})$,
        where $g_{s,s'}(\tilde{\theta}) = (r(s, s') + \gamma \phi(s')^T \tilde{\theta} - \phi(s)^T \tilde{\theta}) \phi(s_t)$.
        $\theta_0 = \tilde{\theta}$.
        **for** $t = 1$ **to** $M$ **do**
            Sample $s, s'$ from $\mathcal{D}$.
            Compute $v_t = g_{s,s'}(\theta_{t-1}) - g_{s,s'}(\tilde{\theta}) + \bar{g}_{m'}(\tilde{\theta})$.
            Update parameters $\theta_t = \theta_{t-1} + \alpha v_t$.
        **end for**
        Set $\tilde{\theta}_{m'} = \theta_{t'}$ for randomly chosen $t' \in (0, \dots, M-1)$.
    **end for**

---

it turns out that for SVRG to work well, we don't need to compute this expectation at every time step, but rather can re-use the computation from a previous iteration. It is proved in Johnson & Zhang (2013) that an optimal frequency of updates between computations of the full update $\mathbb{E}[g_t'(\tilde{\theta})]$ allows the algorithm to achieve a geometric convergence rate (in contrast to SGD, which does not attain a geometric convergence rate).

In this paper we propose a modification of the TD(0) method with SVRG technique (TD-SVRG) which can attain a geometric convergence rate. This algorithm is given above as Algorithm 1. The algorithm works under the "finite sample set" setting which assumes there already exists a sampled data set $\mathcal{D}$. This is the same setting as in Du et al. (2017). However, the method we propose does not add regularization and does not use dual parameters, which makes it considerably simpler.

Like the classic SVRG algorithm, our proposed TD-SVRG has two nested loops. We refer to one step of the outer loop as an *epoch* and to one step of the inner loop as an *iteration*. TD-SVRG keeps two parameter vectors: the current parameter vector $\theta_t$, which is being updated at every iteration, and the vector $\tilde{\theta}_t$, which is updated at the end of each epoch.

Each epoch contains $M$ iterations, which we call update batch size (not to be confused with the estimation batch size, which will be used in the algorithms below to compute an estimate of the mean-path update).

## 4 Outline of the Analysis

In this section we briefly discuss a perspective on TD learning which represents the key difference between our analysis and the previous works. In Xu et al. (2020) the authors note: "In Johnson & Zhang (2013) , the convergence proof relies on the relationship between the gradient and the value of the objective function, but there is not such an objective function in the TD learning problem." We show, that viewing TD learning as gradient splitting allows us to find such a function and establish a relationship between the gradient and the value function.

The concept of viewing TD-learning as gradient splitting comes from Liu & Olshevsky (2021), in which the authors define the linear function $h(\theta) = B(\theta - \theta^*)$ as **gradient splitting** of a quadratic function $f(\theta) = (\theta - \theta^*)^T A(\theta - \theta^*)$ if $B + B^T = 2A$. Liu & Olshevsky define the function:

$$f(\theta) = (1 - \gamma)||V_\theta - V_{\theta^*}||_D^2 + \gamma ||V_\theta - V_{\theta^*}||_{\text{Dir}}^2,$$

where $V_\theta$ is a vector of state values induced by $\theta$,

$$||V||_D^2 = \sum_s \mu_\pi(s) V(s)^2$$

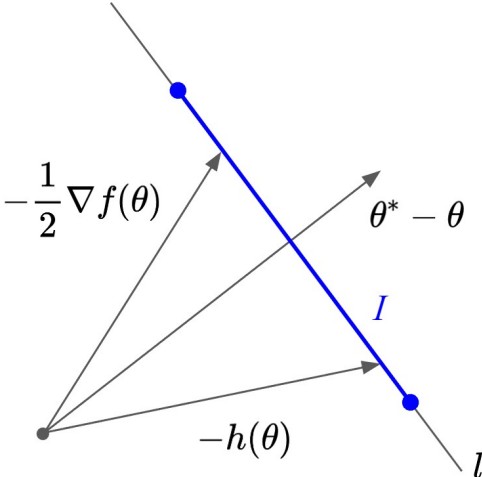

Figure 1: Illustration of gradient splitting. All gradient splittings of the function $f(\theta)$ will lie on line $l$. In addition, if we have a constraint on the 2-norm of the matrix $A$, all gradient splittings will lie on an interval $I$, thus suggesting that an update in the direction of gradient splitting is almost as good, is an update in the direction of the true gradient.

is a weighted norm, and

$$||V||^2_{\text{Dir}} = \frac{1}{2} \sum_{s,s'} \mu_\pi(s) P(s, s')(V(s) - V(s'))^2$$

is a Dirichlet seminorm. They show that mean-path update $-\bar{g}(\theta)$ is a gradient splitting of this function $f(\theta)$, which is how this function naturally appears in the analysis of TD-learning.

Our arguments build on the gradient splitting interpretation of TD, and it is this approach that differentiates our paper from previous works on variance-reduced policy evaluation. This interpretation provides a tool for its convergence analysis, since it leads to bounds on a key quantity: the inner product of a gradient splitting $h(\theta)$ and the direction $\theta^* - \theta$ to the minimizer is the same as the inner product of $-\nabla f(\theta)$ and $\theta^* - \theta$ (see Figure 1). At the same time, gradient splitting is not the gradient itself, and many properties that hold for the gradient do not hold for gradient splitting. Please see Appendix E for additional discussion.

A key difficulty to overcome is that, in the "finite sample" case discussed earlier, the two definitions of the function $f(\theta)$ are no longer equivalent and, as a result, the TD(0) update is no longer a gradient splitting. This complicates things considerably and our key idea is to view TD updates in this case as a form of an approximate gradient splitting.

In addition to $f(\theta)$, we define the expected square norm of the difference between the current and optimal parameters as $w(\theta)$ :

$$w(\theta) \quad = \quad \mathbb{E}[||g_{s,s'}(\theta) - g_{s,s'}(\theta^*)||^2], \tag{3}$$

where expectation is taken with respect to sampled pair of states $s, s'$. With this notation we provide a technical lemma. The next proofs are based on variations of this lemma.

**Lemma 4.1.** *If Assumptions 2.1, 2.2 hold, the epoch parameters of two consecutive epochs $m' - 1$ and $m'$ are related by the following inequality:*

$$2\alpha M \mathbb{E}[f_d(\tilde{\theta}_{m'})] - 2M\alpha^2 \mathbb{E}[w(\tilde{\theta}_{m'})] \le \mathbb{E}[||\tilde{\theta}_{m'-1} - \theta^*||^2] + 2\alpha^2 M \mathbb{E}[w(\tilde{\theta}_{m'-1})], \tag{4}$$

*where the expectation is taken with respect to all previous epochs and choices of states $s, s'$ during the epoch $m$.*

*Proof.* The proof of the lemma generally follows the analysis in Johnson & Zhang (2013) and can be found in Appendix B. □

Lemma 4.1 plays an auxiliary role in our analysis and significantly simplifies it. It introduces a new approach to the convergence proof by carrying iteration to iteration and epoch to epoch bounds to the earlier part of the analysis. In particular, deriving bounds in terms of some arbitrary function $u(\theta)$ is now reduced to deriving upper bounds on $||\tilde{\theta}_{m'-1} - \theta^*||^2$ and $w(\theta)$, and a lower bound on $f(\theta)$ in terms of the function $u$. Three mentioned quantities are natural choices for the function $u$. In Appendix C we show Lemma 4.1 might be used to derive convergence in terms of $||\tilde{\theta}_{m'-1} - \theta^*||^2$ with similar bounds as in Du et al. (2017). In this paper we use $f(\theta)$ as $u$ to improve on previous results.

# 5 Main results

Our main results contain 4 theorems which establish convergence for 4 different settings: TD-SVRG for the finite samples setting (with one extra subseciton which outlines the similarity between the achieved complexity of TD-SVRG and classical SVRG), batched TD-SVRG for the finite sample setting, TD-SVRG for i.i.d. online sampling and TD-SVRG for Markovian online sampling.

## 5.1 Convergence of TD-SVRG for finite sample setting

In this section, we show that Algorithm 1 attains geometric convergence in terms of a specially chosen function $f_d(\theta)$ with $\alpha$ being $\mathcal{O}(1)$ and $M$ being $\mathcal{O}(1/\lambda_A)$. Before we start note that in general a first state of the first pair and a second state of the last state pair in the randomly sampled dataset would not be the same state. That leads to the effect which we call *unbalanced dataset*: unlike the MDP, the first and second states distributions in such a dataset are different. In the unbalanced dataset case, mean path update is not exactly a gradient splitting of the target function $f(\theta)$ and we need to introduce a correction term in our analysis. The following theorem covers the unbalanced dataset case and the balanced dataset case is covered in the corollary.

**Theorem 5.1.** *Suppose Assumptions 2.1, 2.2 hold and the dataset $\mathcal{D}$ may be unbalanced. Define the error term $J = \frac{4\gamma^2}{N\lambda_A}$. Then, if we choose learning rate $\alpha = 1/(8 + J)$ and update batch size $M = 2/(\lambda_A\alpha)$, Algorithm 1 will have a convergence rate of:*

$$\mathbb{E}[f_d(\tilde{\theta}_m)] \leq \left(\frac{2}{3}\right)^m f_d(\tilde{\theta}_0).$$

**Corollary 5.2.** *If the dataset $\mathcal{D}$ is balanced, then we may take the error term is $J = 0$ and consequently the same convergence rate might be obtained with choices of learning rate $\alpha = 1/8$ and update batch size $M = 16/\lambda_A$*

*Proof of Theorem 5.1.* The proof is given in Appendix D. □

Note that $\tilde{\theta}_{m'}$ refers to the iterate after $m$ iterations of the outer loop. Thus, the total number of samples guaranteed by this theorem until $\mathbb{E}[f_d(\tilde{\theta}_m)] \leq \epsilon$ is actually $\mathcal{O}((N + 16/\lambda_A)\log(1/\epsilon))$ in the balanced case and $\mathcal{O}((N + \frac{16+2/(N\lambda_A)}{\lambda_A})\log(1/\epsilon))$ in the unbalanced case, which means that two complexities are identical if the dataset size $N$ is large enough so that $N \geq \lambda_A^{-1}$.

Even an error term introduced by an unbalanced dataset is negligible in the randomly sampled dataset cases; it might not be lower bounded by a value less than $J$. In practice the issue might be tackled by sampling from a modified dataset this issue is discussed in Appendix F.

## 5.2 Similarity of SVRG and TD-SVRG

Note that the dataset case is similar to SVRG in the convex setting in the sense that: 1) the update performed at each step is selected uniformly at random, and 2) the exact mean-path update can be computed at every

epoch. If the dataset is balanced, a negative mean-path update $-\bar{g}(\theta)$ is a gradient splitting of the function $f(\theta)$. These allow us to further demonstrate the significance of the function $f(\theta)$ for the TD learning process and the greater similarity between TD-learning and convex optimization. We recall the convergence rate obtained in Johnson & Zhang (2013) for a sum of convex functions:

$$\frac{1}{\gamma'\alpha'(1-2L\alpha')M'} + \frac{2L\alpha'}{1-2L\alpha'},$$

where $\gamma'$ is a strong convexity parameter and $L$ is a Lipschitz smoothness parameter (we employ the notation from the original paper and introduce the symbol $'$ to avoid duplicates). The function $f(\theta) = \frac{1}{2}(\theta-\theta^*)^T A(\theta-\theta^*)$ is $\lambda_A$ strongly convex and 1-Lipschitz smooth, which means that the convergence rate obtained in this paper is identical to the convergence rate of SVRG in the convex setting. We provide an intuition that supports this similarity in Appendix D. This fact further extends the analogy between TD learning and convex optimization earlier explored by Bhandari et al. (2018) and Liu & Olshevsky (2021).

### 5.3 TD-SVRG with batching

In this section, we extend our results to an inexact mean-path update computation, applying the results of Babanezhad Harikandeh et al. (2015) to the TD SVRG algorithm. We show that the geometric convergence rate might be achieved with a smaller number of computations by estimating the mean-path TD-update instead of performing full computation. This approach is similar to Peng et al. (2020), but does not require dual variables and achieves better results.

Since the computation of the mean-path error is not related to the dataset balance, in this section we assume that the dataset is balanced for simplicity.

**Theorem 5.3.** *Suppose Assumptions 2.1, 2.2 hold and the algorithm runs for a total of $m$ epochs. Then, if the learning rate is chosen as $\alpha = 1/8$, the update batch size is $M = 16/\lambda_A$, and the estimation batch size during epoch $m'$ is $n_{m'} = \min\left(N, \frac{N}{N-1}\frac{1}{c\lambda_A(2/3)^m}(4f(\tilde{\theta}_{m'}) + \sigma^2)\right)$, where $c$ is a parameter and $\sigma^2 = E[g_{s,s'}(\theta^*)]$ is an optimal point update variance, Algorithm 2 will converge to the optimum with a convergence rate of:*

$$\mathbb{E}[f_d(\tilde{\theta}_m)] \leq \left(\frac{2}{3}\right)^m (f_d(\tilde{\theta}_0) + C),$$

*where $C$ is a constant dependent on the parameter $c$.*

*Proof.* The proof is given in Appendix G.1. $\qquad\square$

This result is an improvement on Peng et al. (2020), compared to which it improves both the estimation and update batch sizes. In terms of the update batch size, our result is better by at least a factor of $1/((1-\gamma)\pi_{\min}^3)$, where $\pi_{\min}$ represents the minimum probability within the stationary distribution of the transition matrix, see Table 1 for theoretical results and Section J.4 for experimental comparison. In terms estimation batch size, we have given the result explicitly in terms of the iterate norm, while Peng et al. (2020) has a bound in terms of the variance of both primal and dual update vectors ($\Xi^2$ in their notation).

Note, that both quantities $f(\tilde{\theta}_{m'})$ and $\sigma^2$ required to compute the estimation batch size $n_{m'}$ are not known during the run of the algorithm. However, we provide an alternative quantity, which might be used in practice: $n_{m'} = \min(N, \frac{N}{N-1}\frac{1}{c\lambda_A(2/3)^m}(2|r_{\max}|^2 + 8||\tilde{\theta}_{m'-1}||^2)$, where $|r_{\max}|$ is the maximum absolute reward.

### 5.4 Online i.i.d. sampling from the MDP

We now apply a gradient splitting analysis to TD learning in the case of online i.i.d. sampling from the MDP each time we need to generate a new state $s$. We show that our methods can be applied in this case to derive tighter convergence bounds. One issue of TD-SVRG in the i.i.d. setting is that the mean-path update may not be computed directly. Indeed, once we have a dataset of size $N$, we can simply make a pass through it; but in an MDP setting, it is typical to assume that making a pass through all the states

of the MDP is impossible. The inexactness of mean-path update is addressed with the sampling technique introduced previously in Subsection 5.3, which makes the i.i.d. case very similar to TD-SVRG with non-exact mean-path computation in the finite sample case. Thus, the TD-SVRG algorithm for the i.i.d. sampling case is very similar to Algorithm 2, with the only difference being that states $s, s'$ are being sampled from the MDP instead of the dataset $\mathcal{D}$. Formal description of the algorithm is provided in Appendix H.

In this setting, geometric convergence is not attainable with variance reduction, which always relies on a pass through the dataset. Since here one sample is obtained from the MDP at every step, one needs to use increasing batch sizes. Our algorithm does so, and the next theorem once again improves the scaling with the condition number from quadratic to linear compared to the previous literature.

**Theorem 5.4.** *Suppose Assumptions 2.1, 2.2 hold. Then if the learning rate is chosen as $\alpha = 1/16$, the update batch size as $M = 32/\lambda_A$ and the estimation batch size as $n_{m'} = \frac{1}{c\lambda_A(2/3)^m}(4f(\theta_{m'}) + 2\sigma^2)$, where $c$ is some arbitrary chosen constant, Algorithm 3 will have a convergence rate of:*

$$\mathbb{E}[f_e(\tilde{\theta}_m)] \leq \left(\frac{2}{3}\right)^m (f_e(\tilde{\theta}_0) + C_1),$$

*where $C_1$ is a constant.*

*Proof.* The proof is given in Appendix H. $\qquad\square$

This convergence rate will lead to total computational complexity of $\mathcal{O}(\frac{1}{\lambda_A \epsilon} \log(\epsilon^{-1}))$ to achieve accuracy $\epsilon$.

Similarly to the previous section, a quantity $\frac{1}{c\lambda_A(2/3)^m}(2|r_{max}|^2 + 8||\tilde{\theta}_{m'-1}||^2)$ might be used for estimation batch sizes $n_{m'}$ during practical implementation of the algorithm. Note that the expression $|r_{max}|^2 + 4||\tilde{\theta}_{m'-1}||^2$ is common in the literature, *e.g.*, it is denoted as $D_2$ in Xu et al. (2020).

## 5.5 Online Markovian sampling from the MDP

The Markovian sampling case is the hardest to analyse due to its dependence on the MDP properties, which makes establishing bounds on various quantities used during the proof much harder. Leveraging the gradient splitting view still helps us improve over existing bounds, but the derived algorithm does not have the nice property of a constant learning rate. To deal with sample-to-sample dependencies we introduce one more assumption often used in the literature:

**Assumption 5.5.** For the MDP there exist constants $\bar{m} > 0$ and $\rho \in (0, 1)$ such that

$$\sup_{s \in S} d_{TV}(\mathbb{P}(s_t \in \cdot | s_0 = s), \pi) \leq \bar{m}\rho^t, \quad \forall t \geq 0,$$

where $d_{TV}(P, Q)$ denotes the total-variation distance between the probability measures P and Q.

In the Markovian setting, we also need to employ a projection, which helps to set a bound on the update vector $v$. Following Xu et al. (2020), after each iteration we project the parameter vector on a ball of radius $R$ (denoted as $\Pi_R(\theta) = \arg\min_{\theta':|\theta'| \leq R} |\theta - \theta'|^2$). We assume that $|\theta^*| \leq R$, where the choice of $R$ that satisfies this bound can be found in Section 8.2 at Bhandari et al. (2018). The detailed description of the algorithm is in Appendix I.

**Theorem 5.6.** *Suppose Assumptions 2.1, 2.2, 5.5 hold. Then, the output of Algorithm 4 satisfies:*

$$\mathbb{E}[f_e(\tilde{\theta}_m)] \leq \left(\frac{3}{4}\right)^m f_e(\theta_0) + \frac{8C_2}{\lambda_A n_m} + 4\alpha(2G^2(4 + 6\tau^{\text{mix}}(\alpha)) + 9R^2),$$

*where $C_2 = \frac{4(1+(m-1)\rho)}{(1-\rho)}[4R^2 + r_{\max}^2]$.*

*Proof.* The proof is given in Appendix I. $\qquad\square$

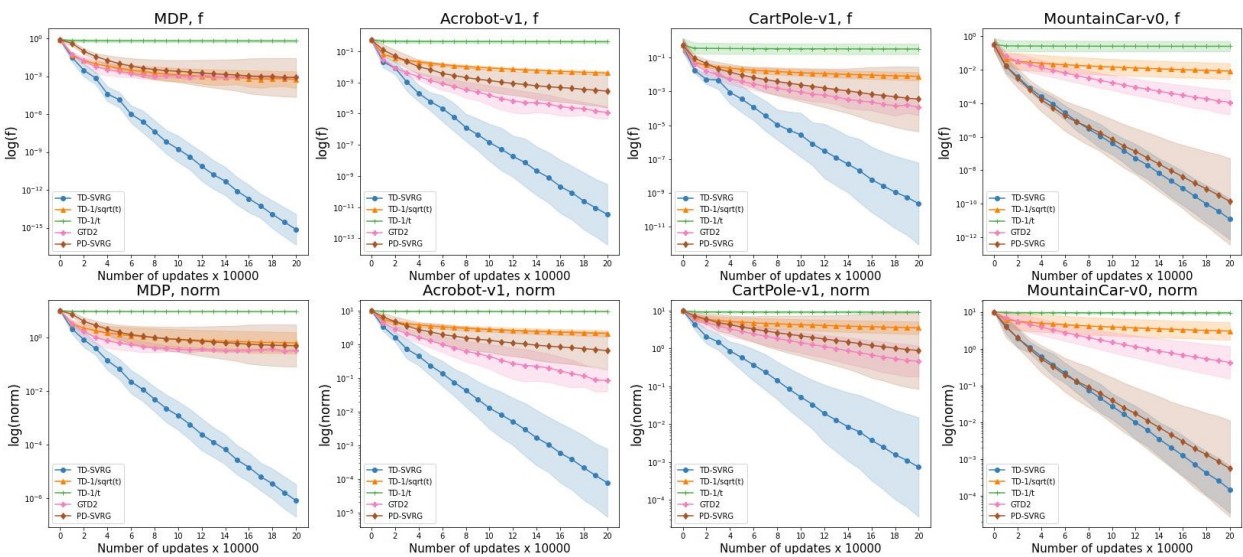

Figure 2: Geometric average performance of different algorithms in the finite sample case. Columns - dataset source environments: MDP, Acrobot, CartPole and Mountain Car. Rows - performance measurements: $\log(f(\theta))$ and $\log(|\theta - \theta^*|)$.

Theorem 5.6 implies that if we choose $s = \mathcal{O}(\log(1/\epsilon))$, $n_{m'} = \mathcal{O}(1/(\lambda_A \epsilon))$, $\alpha = \mathcal{O}(\epsilon/\log(1/\epsilon))$ and $M = \mathcal{O}\left(\frac{\log(1/\epsilon)}{\epsilon \lambda_A}\right)$, the total sample complexity is:

$$\mathcal{O}\left(\frac{\log^2(1/\epsilon)}{\epsilon \lambda_A}\right).$$

This has improved scaling with the condition number $\lambda_A^{-1}$ compared to $\mathcal{O}\left(\frac{1}{\epsilon \lambda_A^2}\log(1/\epsilon)\right)$ in Xu et al. (2020).

## 6 Experimental results

Figure 2 shows the relative performance of TD-SVRG, GTD2 (Sutton et al., 2009), "vanilla" TD learning (Sutton, 1988), and PD-SVRG (Du et al., 2017) in the finite sample setting. We used theory-suggested parameters for TD-SVRG, whereas parameters for PD-SVRG and GTD2 are selected by grid search. Datasets of size 5,000 are generated from 4 environments: Random MDP (Dann et al., 2014), and the Acrobot, CartPole and Mountain car OpenAI Gym environments (Brockman et al., 2016). The complexity (x-axis on the graph) is measured in the number of basic updates computations, which is computing an update $g_{s,s'}(\theta)$ for a sampled pair of states $s, s'$ and parameter vector $\theta$. Note that this complexity accounts for both basic updates required to perform inner loop iterations of the algorithms and updates required to compute or estimate the mean-path update. As the theory predicts, TD-SVRG and PD-SVRG converge geometrically, while GTD and vanilla TD converge sub-linearly.

Details on the experiments and grid search can be found in Appendix J. In addition, Appendix J has more experimental results: comparison of theoretical batch sizes (Appendix J.1), results on a datasets with DQN produced features (Appendix J.3), results for the dataset case with batched estimation of mean-path update (Appendix J.4), parameter search results for TD-SVRG algorithm (Appendix J.2), results of experiments for the online case with i.i.d sampling (Appendix J.5) and Markovian sampling (Appendix J.6). Instructions and code for reproducing the experiments can be found in our github repository.

# 7 Conclusions

In the paper we provide improved sample complexity results for variance-reduced policy evaluation. Our key theoretical finding is that it is possible to reduce the scaling with the condition number of the problem from quadratic to linear, matching what is known for SVRG in the convex optimization setting, while simultaneously removing a number of extraneous factors. This results in a many orders of magnitude improvements for batch size and sample complexity for even simple problems such as random MDPs or OpenAI Gym problems. Results of this type are attained in several settings, e.g., when a dataset of size $N$ is sampled from the MDP, and when states of the MDP are sampled online either in an i.i.d. or Markovian fashion. In simulations we find that our method with step-sizes and batch-sizes coming from our theorems outperforms algorithms from the previous literature with the same parameters selected by grid search. The main innovation in the proofs of our results is to draw on a view of TD learning as an approximate splitting of gradient descent.

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

# A    Discussion on TD-learning lower bound

In this paper, we want to show that the sample efficiency of TD-learning cannot be lower than $\mathcal{O}(\epsilon^{-1})$. This is easy to demonstrate by utilizing the fact that the policy evaluation problem easily reduces to a mean estimation problem, for which we have an established bound of $\mathcal{O}(\epsilon^{-1})$. Here, we give a simple example of how to reduce the policy evaluation problem to the problem of estimating the probability $p$ of a Bernoulli random variable.

Consider a 3 state MDP and a policy which has the following transition matrix between states:

$$P = \begin{pmatrix} 1/2 & p & 1/2 - p \\ 1/2 & 1/2 & 0 \\ 1/2 & 0 & 1/2 \end{pmatrix}.$$

As is standard in policy evaluation, we do not assume the MDP is known; in particular, the parameter $p$ is unknown. The agent receives reward of 1 whenever it moves from state 2 and reward of 0 otherwise. Thus, the reward here depends only on the state. The MDP with this policy is geometrically ergodic since $P^2$ is a strictly positive matrix.

Let us compute the value function assuming we start from state 1. This can be done by observing that the stationary distribution of this probability transition matrix is $(1/2, p, 1/2 - p)$, and that if we start at node 1, it reaches the stationary distribution after a single step. Thus

$$V(1) = 0 + \gamma p \cdot 1 + \gamma^2 p \cdot 1 + \cdots = \gamma \frac{p}{1 - \gamma}.$$

Thus if we take $\gamma = 1/2$, $V(1) = p$ and thus estimating the value function with a certain expected square error will translate to a similar expected square error on estimating $p$, up to constants. Therefore, it is not possible to get the error $\epsilon$ with better complexity than $\mathcal{O}(\epsilon^{-1})$, since it would violate lower bound provided by LeCam's method (Le Cam, 2012).

This simple argument states that the results of i.i.d. case analysis reported directly in Ma et al. (2020) and indirectly implied by Xu et al. (2020) (where complexity better than $\epsilon^{-1}$ might be achieved under certain choices of batch size and learning rate) are not possible to achieve.

## B    Proof of Lemma 4.1

The proof follows the same logic as in Johnson & Zhang (2013) and is organized in four steps.

*Step* B.1. In the original paper, the proof starts with deriving a bound on the squared norm of the difference between the current and optimal parameter vectors. With the introduction of $w(\theta)$ this step in our proof is trivial. We have

$$\mathbb{E}_{s,s'}||g_{s,s'}(\theta) - g_{s,s'}(\theta^*)||^2 = w(\theta),$$

where $\mathbb{E}_{s,s'}$ denotes the expectation taken with respect to the choice of a random pair of states $s, s'$. In other words, $\mathbb{E}_{s,s'}[\cdot]$ denotes the conditional expectation with respect to all variables that are not $s, s'$, which, recall, are generated at time $t$ by sampling $s$ from the stationary distribution and letting $s'$ be the next state. We will slightly abuse notation to write $\mathbb{E}_{s,s'}[\cdot]$ instead of the more rigorous $\mathbb{E}_{s_t,s_{t+1}}$, since what time index the states are generated at random is usually clear from the context.

*Step* B.2. During Step 2 we derive a bound on the norm of a single iteration $t$ update $v_t = g_{s,s'}(\theta_{t-1}) - g_{s,s'}(\tilde{\theta}) + \bar{g}(\tilde{\theta})$, where $\bar{g}(\tilde{\theta})$ is defined in 1 assuming that states $s, s'$ were sampled randomly during step $t$:

$$
\begin{aligned}
\mathbb{E}_{s,s'}[||v_t||^2] &= \mathbb{E}_{s,s'}||g_{s,s'}(\theta_{t-1}) - g_{s,s'}(\tilde{\theta}) + \bar{g}(\tilde{\theta})||^2 \\
&= \mathbb{E}_{s,s'}||(g_{s,s'}(\theta_{t-1}) - g_{s,s'}(\theta^*)) + (g_{s,s'}(\theta^*) - g_{s,s'}(\tilde{\theta}) + \bar{g}(\tilde{\theta})||^2 \\
&\leq 2\mathbb{E}_{s,s'}||(g_{s,s'}(\theta_{t-1}) - g_{s,s'}(\theta^*))||^2 \\
&\quad + 2\mathbb{E}_{s,s'}||g_{s,s'}(\tilde{\theta}) - g_{s,s'}(\theta^*) - (\bar{g}(\tilde{\theta}) - \bar{g}(\theta^*))||^2 \\
&= 2\mathbb{E}_{s,s'}||(g_{s,s'}(\theta_{t-1}) - g_{s,s'}(\theta^*))||^2 + 2\mathbb{E}_{s,s'}||g_{s,s'}(\tilde{\theta}) - g_{s,s'}(\theta^*) \\
&\quad - \mathbb{E}_{s,s'}[g_{s,s'}(\tilde{\theta}) - g_{s,s'}(\theta^*)]||^2 \\
&\leq 2\mathbb{E}_{s,s'}||(g_{s,s'}(\theta_{t-1}) - g_{s,s'}(\theta^*))||^2 + 2\mathbb{E}_{s,s'}||g_{s,s'}(\tilde{\theta}) - g_{s,s'}(\theta^*)||^2 \\
&= 2w(\theta_{t-1}) + 2w(\tilde{\theta}).
\end{aligned}
$$

The first inequality uses $\mathbb{E}||a + b||^2 \leq 2\mathbb{E}||a||^2 + 2\mathbb{E}||b||^2$. The second inequality uses the fact that the second central moment is smaller than the second moment. The last equality uses the equality from Step 1.

*Step* B.3. During this step we derive a bound on the expected squared norm of a distance to the optimal parameter vector after a single update $t$:

$$
\begin{aligned}
\mathbb{E}_{s,s'}||\theta_t - \theta^*||^2 &= \mathbb{E}_{s,s'}||\theta_{t-1} - \theta^* + \alpha v_t||^2 \\
&= ||\theta_{t-1} - \theta^*||^2 + 2\alpha(\theta_{t-1} - \theta^*)^T \mathbb{E}_{s,s'} v_t + \alpha^2 \mathbb{E}_{s,s'}||v_t||^2 \\
&\leq ||\theta_{t-1} - \theta^*||^2 + 2\alpha(\theta_{t-1} - \theta^*)^T \bar{g}(\theta_{t-1}) + 2\alpha^2 w(\theta_{t-1}) + 2\alpha^2 w(\tilde{\theta}) \\
&= ||\theta_{t-1} - \theta^*||^2 - 2\alpha f_d(\theta_{t-1}) + 2\alpha^2 w(\theta_{t-1}) + 2\alpha^2 w(\tilde{\theta}).
\end{aligned}
$$

The inequality uses the bound obtained in Step 2 and equality uses gradient splitting properties of $\bar{g}(\theta_{t-1})$:

$$
\begin{aligned}
(\theta_{t-1} - \theta^*)^T \bar{g}(\theta_{t-1}) &= (\theta_{t-1} - \theta^*)^T (\bar{g}(\theta_{t-1}) - \bar{g}(\theta^*)) \\
&= (\theta_{t-1} - \theta^*)^T (-A_d \theta_{t-1} + b + A_d \theta^* - b) \\
&= -(\theta_{t-1} - \theta^*)^T A_d (\theta_{t-1} - \theta^*) = -f_d(\theta_{t-1}).
\end{aligned}
\tag{5}
$$

After rearranging terms it becomes:

$$\mathbb{E}_{s,s'}||\theta_t - \theta^*||^2 + 2\alpha f_d(\theta_{t-1}) - 2\alpha^2 w(\theta_{t-1}) \leq ||\theta_{t-1} - \theta^*||^2 + 2\alpha^2 w(\tilde{\theta}).$$

*Step* B.4. During this step we sum the inequality obtained in Step 3 over the epoch and take another expectation to obtain:

$$\mathbb{E}[\sum_{t=1}^{M}||\theta_t - \theta^*||^2 + \sum_{t=1}^{M} 2\alpha \mathbb{E} f_d(\theta_{t-1}) - \sum_{t=1}^{M} 2\alpha^2 w(\theta_{t-1})|\mathcal{F}_{m'-1}] \leq \mathbb{E}[\sum_{t=1}^{M}||\theta_{t-1} - \theta^*||^2 + \sum_{t=1}^{M} 2\alpha^2 w(\tilde{\theta})|\mathcal{F}_{m'-1}], \tag{6}$$

where $\mathcal{F}_{m'-1}$ is the information available in the beginning of epoch $m'$. We analyze this expression term-wise.

Notice that $\sum_{t=1}^{M} ||\theta_{t-1} - \theta^*||^2$ and $\sum_{t=1}^{M} ||\theta_t - \theta^*||^2$ consist of the same terms, except the first term in the first sum and the last term in the last sum, which are $||\theta_0 - \theta^*||^2$ and $||\theta_M - \theta^*||^2$ respectively. Since $||\theta_M - \theta^*||^2$ is always positive and it is on the left hand side of the inequality, we could drop it.

We denote the parameter vector $\theta$ chosen for epoch parameters at the end of the epoch $\tilde{\theta}_{m'}$. Since this vector is chosen uniformly at random among all iteration vectors $\theta_t$, $t \in (0, M-1)$, we have that $\sum_{t=1}^{M} \mathbb{E}f_d(\theta_{t-1}) = M\mathbb{E}f_d(\tilde{\theta}_{m'})$ and $\sum_{t=1}^{M} \mathbb{E}w(\theta_{t-1}) = M\mathbb{E}w(\tilde{\theta}_{m'})$.

At the same time, $\tilde{\theta}$, which was chosen at the end of the previous epoch remains the same throughout the epoch, therefore, $\sum_{t=1}^{M} \mathbb{E}w(\tilde{\theta}) = M\mathbb{E}w(\tilde{\theta})$. Note, that the current epoch starts with setting $\theta_0 = \tilde{\theta}$. Also, to underline that $\tilde{\theta}$ during the current epoch refers to the previous epoch, we denote it as $\tilde{\theta}_{m'-1}$. Plugging these values in (4) we have :

$$2\alpha M\mathbb{E}f_d(\tilde{\theta}_{m'}) - 2M\alpha^2\mathbb{E}w(\tilde{\theta}_{m'}) \leq \mathbb{E}||\tilde{\theta}_{m'-1} - \theta^*||^2 + 2\alpha^2 M\mathbb{E}w(\tilde{\theta}_{m'-1}).$$

## C  Convergence in terms of squared norm

At this point, we go on an aside to prove a result that is not in the main body of the paper. We observe it is possible to derive a bound on Algorithm 1 in the squared norm. This bound is generally worse than the results we report in the main body of the paper since it scales with the square of the condition number.

**Proposition C.1.** *Suppose Assumptions 2.1, 2.2 hold. If we chose the learning rate as $\alpha = \lambda_A/32$ and update batch size as $M = 32/\lambda_A^2$, then Algorithm 1 has a convergence rate of:*

$$\mathbb{E}[||\tilde{\theta}_{m'} - \theta^*||^2] \leq \left(\frac{5}{7}\right)^m ||\tilde{\theta}_0 - \theta^*||^2.$$

This leads to batch size $M$ being $\mathcal{O}(1/\lambda_A^2)$, which is better than the results in Du et al. (2017), since their results have complexity $\mathcal{O}(\kappa^2(C)\kappa_G^2)$, where $\kappa(C)$ is the condition number of matrix $C = \mathbb{E}_{s\in\mathcal{D}}[\phi(s)\phi(s)^T]$ and $\kappa_G \propto 1/\lambda_{\min}(A^T C^{-1} A)$.

*Proof.* To transform inequality (4) from Lemma 4.1 into a convergence rate guarantee, we need to bound $w(\theta)$ and $f_d(\theta)$ in terms of $||\theta - \theta^*||^2$. Both bounds are easy to show:

$$
\begin{aligned}
w(\theta) &= \mathbb{E}_{s,s'}||g_{s,s'}(\theta) - g_{s,s'}(\theta^*)||^2 \\
&= (\theta - \theta^*)^T \mathbb{E}_{s,s'}[(\gamma\phi(s') - \phi(s))\phi(s)^T \phi(s)(\gamma\phi(s') - \phi(s))^T](\theta - \theta^*) \\
&\leq (\theta - \theta^*)^T \mathbb{E}_{s,s'}[||(\gamma\phi(s') - \phi(s))|| \cdot ||\phi(s)|| \cdot ||\phi(s)|| \cdot ||(\gamma\phi(s') - \phi(s))||](\theta - \theta^*) \\
&\leq 4||\theta - \theta^*||^2, \\
f_d(\theta) &= (\theta - \theta^*)^T \mathbb{E}_{s,s'}[\phi(s)(\phi(s) - \gamma\phi(s'))^T](\theta - \theta^*) \geq \lambda_A ||\theta - \theta^*||^2,
\end{aligned}
$$

where $\mathbb{E}_{s,s'}$ denotes the expectation taken with respect to a choice of pair of states $s, s'$. Plugging these bounds into Equation (4) we have:

$$(2\alpha M\lambda_A - 8M\alpha^2)||\tilde{\theta}_{m'} - \theta^*||^2 \leq (1 + 8M\alpha^2)||\tilde{\theta}_{m'-1} - \theta^*||^2,$$

which yields an epoch to epoch convergence rate of:

$$\frac{1 + 8M\alpha^2}{2\alpha M\lambda_A - 8M\alpha^2}.$$

For this expression to be $< 1$, we need that $\alpha M$ is set to $\mathcal{O}(1/\lambda_A)$, which means that $\alpha$ needs to be $\mathcal{O}(\lambda_A)$ for $M\alpha^2$ to be $\mathcal{O}(1)$. Therefore, $M$ needs to be $\mathcal{O}(1/\lambda_A^2)$. Setting $\alpha = \lambda_A/32$ and $M = 32/\lambda_A^2$ yields a convergence rate of 5/7. □

# D Proof of Theorem 5.1

An analysis of the balanced dataset case follows from unbalanced dataset but for clarity of presentation we provide a proof for balanced dataset separately, but before diving into it, let us provide an intuition as to why TD-SVRG in this case exhibits the same convergence as in the convex optimization case.

Let's assume we are solving a convex optimization problem for the function $f_d(\theta)$, *i.e.*, we have access to the true gradients of the functions $f_{s,s'}$. In this case the update at time $t$ is

$$g'_t = \frac{1}{2}\nabla f_t(\theta_t) = \frac{1}{2}(A_t + A_t^T)(\theta - \theta^*).$$

In this case, the results of the SVRG paper are directly applicable. Instead, in the TD setting, we have updates of the form:

$$g_t = A_t\theta_t + b_t.$$

We can see that the TD update is quite different from the convex update, as it has a different linear function and an extra term $b_t$, which would affect the convergence as extra noise. However, once we apply the SVRG technique to these updates, as described in Section 3, the new updates become

$$\begin{aligned} v'_t &= \frac{1}{2}\nabla f_t(\theta_t) - \frac{1}{2}\nabla f_t(\tilde{\theta}) + \mathbb{E}_{s,s'}[\frac{1}{2}\nabla f_{s,s'}(\tilde{\theta})] \\ &= \frac{1}{2}(A_t + A_t^T)(\theta_t - \tilde{\theta}) + \frac{1}{2n}\sum_{s,s'}(A_{s,s'} + A_{s,s'}^T)(\tilde{\theta} - \theta^*) \end{aligned}$$

in the convex case and

$$\begin{aligned} v_t &= (A_t\theta_t + b_t) - (A_t\tilde{\theta} + b_t) + \mathbb{E}_{s,s'}[A_{s,s'}\tilde{\theta} + b_{s,s'}] \\ &= A_t(\theta_t - \tilde{\theta}) + \frac{1}{n}\sum_{s,s'} A_{s,s'}(\tilde{\theta} - \theta^*), \end{aligned}$$

in the TD case, where we again use the fact $\mathbb{E}_{s,s'}[b_{s,s'}] = \mathbb{E}_{s,s'}[-A_{s,s'}\theta^*]$ to establish the equality.

The two updates look much more similar after applying the SVRG technique to them since the extra "noise" term $b_t$ gets canceled with probability 1. Also, $v_t$ is a splitting of the true gradient $v'_t$, which suggests that the application of $v_t$ updates instead of $v'_t$ updates results in the same convergence rate. The formal proof of this fact is given below.

## D.1 Balanced dataset case

Similar to the previous section, we start with deriving bounds, but this time we bound $||\theta - \theta^*||^2$ and $w(\theta)$ in terms of $f_d(\theta)$. The first bound is straightforward:

$$f_d(\theta) = (\theta - \theta^*)^T \mathbb{E}_{s,s'}[\phi(s)(\phi(s) - \gamma\phi(s')^T](\theta - \theta^*) \implies ||\theta - \theta^*||^2 \leq \frac{1}{\lambda_A} f_d(\theta),$$

where $\mathbb{E}_{s,s'}$ denotes the expectation taken with respect to a choice of pair of states $s, s'$. For $w(\theta)$ we have:

$$
\begin{aligned}
w(\theta) &= (\theta - \theta^*)^T \mathbb{E}_{s,s'}[(\gamma\phi(s') - \phi(s))\phi(s)^T\phi(s)(\gamma\phi(s') - \phi(s))^T](\theta - \theta^*) \\
&= (\theta - \theta^*)^T \Big[\frac{1}{N}\sum_{s,s'\in\mathcal{D}}(\gamma\phi(s') - \phi(s))\phi^T(s)\phi(s)(\gamma\phi(s') - \phi(s))^T\Big](\theta - \theta^*) \\
&\leq (\theta - \theta^*)^T \Big[\frac{1}{N}\sum_{s,s'\in\mathcal{D}}(\gamma\phi(s') - \phi(s))(\gamma\phi(s') - \phi(s))^T\Big](\theta - \theta^*) \\
&= (\theta - \theta^*)^T \Big[\frac{1}{N}\sum_{s,s'\in\mathcal{D}}\gamma^2\phi(s')\phi(s')^T - \gamma\phi(s')\phi(s)^T\Big](\theta - \theta^*) + f_d(\theta) \\
&= (\theta - \theta^*)^T \Big[\frac{1}{N}\sum_{s,s'\in\mathcal{D}}\gamma^2\phi(s)\phi(s)^T - \gamma\phi(s)\phi(s')^T\Big](\theta - \theta^*) + f_d(\theta) \\
&\leq 2f_d(\theta),
\end{aligned}
\tag{7}
$$

where the first inequality uses Assumption 2.2, the third equality uses the dataset balance property, and $\sum_{s'}\gamma^2\phi(s')\phi(s')^T = \sum_s \gamma^2\phi(s)\phi(s)^T$, since $s$ and $s'$ are the same set of states. The last inequality uses the fact that $\gamma < 1$.

Plugging these bounds into Equation (4), we have:

$$
2\alpha M\mathbb{E}f_d(\tilde\theta_{m'}) - 4M\alpha^2\mathbb{E}f_d(\tilde\theta_{m'}) \leq \frac{1}{\lambda_A}\mathbb{E}f_d(\tilde\theta_{m'-1}) + 4\alpha^2 M\mathbb{E}f_d(\tilde\theta_{m'-1}),
$$

which yields an epoch to epoch convergence rate of:

$$
\mathbb{E}f_d(\tilde\theta_{m'}) \leq \Big[\frac{1}{2\lambda_A\alpha M(1-2\alpha)} + \frac{2\alpha}{1-2\alpha}\Big]\mathbb{E}f_d(\tilde\theta_{m'-1}).
$$

Setting $\alpha = \frac{1}{8}$ and $M = \frac{16}{\lambda_A}$ we have the desired inequality.

### D.2 Unbalanced dataset case

To prove the theorem we follow the same strategy as in D. For the $f_d(\theta)$ we can use the same bound:

$$
f_d(\theta) = (\theta - \theta^*)^T E_{s,s'}[\phi(s)(\phi(s) - \gamma\phi(s')^T](\theta - \theta^*) \implies ||\theta - \theta^*||^2 \leq \frac{1}{\lambda_A}f_d(\theta).
$$

The bound for $w(\theta)$ is a little bit more difficult:

$$
\begin{aligned}
w(\theta) &= (\theta - \theta^*)^T\Big[\frac{1}{N}\sum_{s,s'\in\mathcal{D}}(\gamma\phi(s') - \phi(s))\phi^T(s)\phi(s)(\gamma\phi(s') - \phi(s))^T\Big](\theta - \theta^*) \\
&\leq (\theta - \theta^*)^T\Big[\frac{1}{N}\sum_{s,s'\in\mathcal{D}}(\gamma\phi(s') - \phi(s))(\gamma\phi(s') - \phi(s))^T\Big](\theta - \theta^*) \\
&= (\theta - \theta^*)^T\Big[\frac{1}{N}\sum_{s,s'\in\mathcal{D}}\gamma\phi(s')(\gamma\phi(s') - \phi(s))^T - \phi(s)(\gamma\phi(s') - \phi(s))^T\Big](\theta - \theta^*) \\
&= (\theta - \theta^*)^T\Big[\frac{1}{N}\sum_{s,s'\in\mathcal{D}}\gamma^2\phi(s')\phi(s')^T - \gamma\phi(s')\phi(s)^T\Big](\theta - \theta^*) + f_d(\theta) \\
&= (\theta - \theta^*)^T\Big[\frac{1}{N}\sum_{s,s'\in\mathcal{D}}\gamma^2\phi(s)\phi(s)^T - \gamma\phi(s)\phi(s')^T\Big](\theta - \theta^*) + f_d(\theta) \\
&\quad + \frac{\gamma^2}{N}(\theta - \theta^*)^T(\phi(s_{N+1})\phi(s_{N+1})^T - \phi(s_1)\phi(s_1)^T)(\theta - \theta^*)^T \\
&\leq 2f_d(\theta) + \frac{\gamma^2}{N}(\theta - \theta^*)^T(\phi(s_{N+1})\phi(s_{N+1})^T - \phi(s_1)\phi(s_1)^T)(\theta - \theta^*)^T.
\end{aligned}
$$

The first inequality follows from Assumption 2.2. The third equality is obtained by adding and subtracting $\frac{\gamma^2}{N}(\theta - \theta^*)^T \phi(s_1)\phi(s_1)^T(\theta - \theta^*)$. The second inequality uses the fact that $\gamma^2 < 1$. We denote the maximum eigenvalue of the matrix $\phi(s_{N+1})\phi(s_{N+1})^T - \phi(s_1)\phi(s_1)^T$ by $\mathcal{K}$ (note that $\mathcal{K} \leq 1$). Thus,

$$w(\theta) \leq 2f_d(\theta) + \frac{\gamma^2 \mathcal{K}}{N}||\theta - \theta^*||^2 \leq f_d(\theta)(2 + \frac{\gamma^2 \mathcal{K}}{N\lambda_A}) \leq f_d(\theta)(2 + \frac{\gamma^2}{N\lambda_A}).$$

Plugging these bounds into Equation (4) we have:

$$(2\alpha M - 2M\alpha^2(2 + \frac{\gamma^2}{N\lambda_A}))\mathbb{E}f_d(\tilde{\theta}_{m'}) \leq (\frac{1}{\lambda_A} + 2\alpha^2 M(2 + \frac{\gamma^2}{N\lambda_A}))f_d(\tilde{\theta}_{m'-1}),$$

which yields a convergence rate of:

$$\frac{1}{\lambda_A 2\alpha M(1 - \alpha(2 + \frac{\gamma^2}{N\lambda_A}))} + \frac{\alpha(2 + \frac{\gamma^2}{N\lambda_A})}{1 - \alpha(2 + \frac{\gamma^2}{N\lambda_A})}.$$

To achieve constant convergence rate, for example $\frac{2}{3}$, we set up $\alpha$ such that $\alpha(2 + \frac{\gamma^2}{N\lambda_A}) = 0.25$, thus the second term is equal to $1/3$ and $\alpha = \frac{1}{8 + \frac{4\gamma^2}{N\lambda_A}}$. Then, to make the first term equal to $1/3$, we need to set

$$M = \frac{2}{\lambda_A \alpha} = \frac{2}{\lambda_A \frac{1}{8 + \frac{4\gamma^2}{N\lambda_A}}}.$$

Thus, $\alpha$ is on the order of $\frac{1}{\max(1, 1/(N\lambda_A))})$ and $M$ is on the order of $\frac{1}{\lambda_A \min(1, N\lambda_A)}$.

# E   Properties of gradient splitting

While gradient splitting is one of our main tools, it is not true that this interpretation can be simply used to carry over results from convex optimization to policy evaluation. To illustrate this point, consider the following properties of convex functions with $L$-smooth gradients:

1. $||\nabla f(x)|| \leq L||x - x^*||^2,$

2. $f(x) - f(x^*) \leq \nabla f(x)^T(x - x^*),$

3. $f(y) \geq f(x) + \nabla f(x)^T(y - x),$

4. $||\nabla f(x) - \nabla f(y)||^2 \leq L(\nabla f(x) - \nabla f(y))^T(x - y),$

5. $f(y) \leq f(x) + \nabla f(x)^T(y - x) + \frac{L}{2}||y - x||^2,$

6. $\alpha f(x) + (1 - \alpha)f(y) \leq \alpha f(x) + (1 - \alpha)f(y) - (\alpha(1 - \alpha)/(2L))||\nabla f(x) - \nabla f(y)||^2,$

7. $f(y) \geq f(x) + \nabla f(x)^T(y - x) + \frac{1}{2L}||\nabla f(x) - \nabla f(y)||^2.$

Now consider the following question: suppose we replace each instance of a gradient by gradient splitting; which of the above inequalities still hold? It turns out that (2), (4), (6) still work with gradient splittings, but (1), (3), (5), (7) do not.

Proofs in the convex optimization literature will typically use some subset of the inequalities (1)-(7), and when porting these arguments to the convex optimization literature, they must be reworked to use only (2), (4), (6). Sometimes this will be trivial, but sometimes this may require a lot of creativity. Adopting the proofs to use gradient splitting instead of the gradient is one of the technical contributions of this paper.

## F    Discussion on Unbalanced dataset

If the dataset balance assumption is not satisfied, it is always possible to modify the MDP slightly and make it satisfied. Indeed, suppose we are given an MDP $M$ with initial state (or distribution) $s_0$ and discount factor $\gamma$. We can then modify the transition probabilities by always transitioning to $s_0$ with probability $p$ regardless of state and action chosen (and doing the normal transition from the MDP $M$ with probability $1 - p$), and changing the discount factor to a new $\gamma'$. Calling the new MDP $M'$, we have that:

- It is very easy to draw a dataset from $M'$ such that the last state is the same as the first one (just make sure to end on a transition to $s_0$!) and the collected dataset will have the dataset balance property.

- Under appropriate choice of $p$ and $\gamma'$, the value function $V_M$ in the original MDP can be easily recovered from the value function of the new MDP $V_{M'}$.

A formal statement of this is in the comment below. Note that all we need to be able to do is change the discount factor (which we usually set) as well as be able to restart the MDP (which we can do in any computer simulation).

The only caveat that the size of the dataset one can draw this way will have to be at least $(1 - \gamma)^{-1}$ in expectation because to make the above sketch work will require a choice of $p$ that is essentially proportional to $(1 - \gamma)$ (see Theorem statement in the next comment for a formal statement). This is not a problem in practice, as typical discount factors are usually $\approx 0.99$, whereas datasets tend to be many orders of magnitude bigger than $\approx 100 = (1 - \gamma)^{-1}$. Even a discount factor of $\approx 0.999$, much closer to one than is used in practice, only forces us to draw a dataset of size 1000 in expectation.

**Theorem F.1.** *Choose*

$$\gamma' = \frac{1 + \gamma}{2}, \quad p = \frac{1 - \gamma}{1 + \gamma},$$

*and consider the pair of MDPs $M$ and $M'$ which are defined in our previous comment. Then the quantities $V_M(s)$ and $V_{M'}(s)$ satisfy the following recursion:*

$$V_M(s) = V_{M'}(s) + \frac{\gamma(1 - \gamma)}{1 + \gamma - 2\gamma^2} V_{M'}(s_0)$$

*Proof.* Let $T$ denote be a time step when the first reset appears. We can condition on $T$ to represent $V_{M'}(s)$ as:

$$V_{M'}(s) = \sum_{t'=1}^{\infty} P(t' = T) E[V_{M'}(s)|t' = T]$$

$$= \sum_{t'=1}^{\infty} (1 - p)^{t'-1} p \left( \left( \sum_{t=1}^{t'} \gamma'^{t-1} E[r_t] \right) + \gamma'^{t'} V_{M'}(s_0) \right),$$

where the expected rewards $E[r_t]$ are the same as in the original MDP. We next change the order of summations:

$$V_{M'}(s) = \sum_{t=1}^{\infty} \left( \gamma'^{t-1} E[r(t)] \sum_{t'=t}^{\infty} (1 - p)^{t'-1} p \right) + \sum_{t'=1}^{\infty} (1 - p)^{t'-1} p \gamma'^{t'} V_{M'}(s_0)$$

$$= \sum_{t'=1}^{\infty} \gamma'^{t'-1} (1 - p)^{t'-1} E[r(t)] + (1 - p)^{t'-1} p \gamma'^{t'} V_{M'}(s_0).$$

Now we use the fact that the chosen $\gamma' = \gamma/(1-p)$ and perform some algebraic manipulations:

$$V_{M'}(s) = \sum_{t'=1}^{\infty} \gamma^{t'-1} E[r(t)] + (1-p)^{t'-1} p \gamma'^{t'} V_{M'}(s_0)$$

$$= V_M(s) + \sum_{t'=1}^{\infty} (1-p)^{t'-1} p \gamma'^{t'} V_{M'}(s_0)$$

$$= V_M(s) + \frac{\gamma p}{1-\gamma p} V_{M'}(s_0),$$

which implies the claimed equality:

$$V_M(s) = V_{M'}(s) - \frac{\gamma(1-\gamma)}{1+\gamma-2\gamma^2} V_{M'}(s_0)$$

$\square$

As claimed above, this theorem can be used to recover $V_M$ from $V_{M'}$. Consequently, the artificial addition of a reset button as above makes it possible to generate a dataset which satisfies our dataset balance assumption from any MDP.

# G TD-SVRG with batching exact algorithm and proof

The algorithm for the batching case is given as follows:

---

**Algorithm 2** TD-SVRG with batching for the finite sample case

---

**Parameters** update batch size $M$ and learning rate $\alpha$.
**Initialize** $\tilde{\theta}_0$.
**for** $m' = 1, 2, ..., m$ **do**
  $\tilde{\theta} = \tilde{\theta}_{m'-1}$,
  choose estimation batch size $n_{m'}$,
  sample batch $\mathcal{D}^{m'}$ of size $n_{m'}$ from $\mathcal{D}$ w/o replacement,
  compute $g_{m'}(\tilde{\theta}) = \frac{1}{n_{m'}} \sum_{s,s' \in \mathcal{D}^{m'}} g_{s,s'}(\tilde{\theta})$,
  where $g_{s,s'}(\tilde{\theta}) = (r(s,s') + \gamma \phi(s')^T \tilde{\theta} - \phi(s)^T \tilde{\theta})\phi(s_t)$.
  $\theta_0 = \tilde{\theta}$.
  **for** $t = 1$ **to** $M$ **do**
    Sample $s, s'$ from $\mathcal{D}$.
    Compute $v_t = g_{s,s'}(\theta_{t-1}) - g_{s,s'}(\tilde{\theta}) + g_{m'}(\tilde{\theta})$.
    Update parameters $\theta_t = \theta_{t-1} + \alpha v_t$.
  **end for**
  Set $\tilde{\theta}_{m'} = \theta_{t'}$ for randomly chosen $t' \in (0, \ldots, M-1)$.
**end for**

---

## G.1 Proof of Theorem 5.3

In the first part of the proof we derive an inequality which relates model parameters of two consecutive epochs similar to what we achieved in previous proofs, but now we introduce error vector to show that the mean path update is estimated instead of being computed exactly. In this proof, we follow the same 4 steps we introduced in the proof of Lemma 4.1. In the second part of the proof we show that there are conditions under which the error term converges to 0.

*Step* G.1. During the first step we use the bound obtained in inequality (7):

$$w(\theta) \leq 2 f_d(\theta).$$

*Step* G.2. During this step we derive a bound on the squared norm of a single update $\mathbb{E}[||v_t||^2]$. But now, compared to previous case, we do not compute the exact mean-path updated $\bar{g}(\theta)$, but its estimate, and assume our computation has error $g_{m'}(\theta) = \bar{g}(\theta) + \eta_{m'}$. Thus, during iteration $t$ of epoch $m$ the single update vector is

$$v_t = g_t(\theta_{t-1}) - g_t(\tilde{\theta}) + \bar{g}(\tilde{\theta}) + \eta_{m'}.$$

Taking expectation conditioned on all history previous to epoch $m$, which we denote as $\mathcal{F}_{m'-1}$, the bound on the single update can be derived as:

$$
\begin{aligned}
\mathbb{E}[||v_t||^2|\mathcal{F}_{m'-1}] &= \mathbb{E}[||g_t(\theta_{t-1}) - g_t(\tilde{\theta}) + \bar{g}(\tilde{\theta}) + \eta_{m'}||^2|\mathcal{F}_{m'-1}] \\
&= \mathbb{E}[||(g_t(\theta_{t-1}) - \bar{g}(\theta^*)) + (\bar{g}(\theta^*) - g_t(\tilde{\theta}) + \bar{g}(\tilde{\theta}) + \eta_{m'})||^2|\mathcal{F}_{m'-1}] \\
&\leq 2\mathbb{E}[||(g_t(\theta_{t-1}) - g_t(\theta^*))||^2|\mathcal{F}_{m'-1}] + \\
&\quad 2\mathbb{E}[||g_t(\tilde{\theta}) - g_t(\theta^*) - (\bar{g}(\tilde{\theta}) - \bar{g}(\theta^*)) - \eta_{m'}||^2|\mathcal{F}_{m'-1}] \\
&= 2\mathbb{E}[||(g_t(\theta_{t-1}) - g_t(\theta^*))||^2|\mathcal{F}_{m'-1}] + \\
&\quad 2\mathbb{E}[||g_t(\tilde{\theta}) - g_t(\theta^*) - \mathbb{E}[g_t(\tilde{\theta}) - g_t(\theta^*)] - \eta_{m'}||^2|\mathcal{F}_{m'-1}] \\
&= 2\mathbb{E}[||(g_t(\theta_{t-1}) - g_t(\theta^*))||^2|\mathcal{F}_{m'-1}] + \\
&\quad 2\mathbb{E}[||g_t(\tilde{\theta}) - g_t(\theta^*) - E[g_t(\tilde{\theta}) - g_t(\theta^*)]||^2|\mathcal{F}_{m'-1}] \\
&\quad - 4\mathbb{E}[\langle g_t(\tilde{\theta}) - g_t(\theta^*) - E[g_t(\tilde{\theta}) - g_t(\theta^*)], \eta_{m'}\rangle|\mathcal{F}_{m'-1}] + 2\mathbb{E}[||\eta_{m'}||^2|\mathcal{F}_{m'-1}] \\
&\leq 2\mathbb{E}[||(g_t(\theta_{t-1}) - g_t(\theta^*))||^2|\mathcal{F}_{m'-1}] + \\
&\quad 2\mathbb{E}[||g_t(\tilde{\theta}) - g_t(\theta^*)||^2|\mathcal{F}_{m'-1}] + 2\mathbb{E}[||\eta_{m'}||^2|\mathcal{F}_{m'-1}] \\
&= 2\mathbb{E}[w(\theta_{t-1}) + 2w(\tilde{\theta}) + 2||\eta_{m'}||^2|\mathcal{F}_{m'-1}] \\
&\leq \mathbb{E}[4f_d(\theta_{t-1}) + 4f_d(\tilde{\theta}) + 2\mathbb{E}||\eta_{m'}||^2|\mathcal{F}_{m'-1}],
\end{aligned}
$$

where the first inequality uses $\mathbb{E}||A + B||^2 \leq 2\mathbb{E}||A||^2 + 2\mathbb{E}||B||^2$, the second inequality uses $\mathbb{E}||A - \mathbb{E}[A]||^2 \leq \mathbb{E}||A||^2$ and the fact $\mathbb{E}[\eta_{m'}|g(\tilde{\theta}) - g(\theta^*) - \mathbb{E}_{s,s'}[g(\tilde{\theta}) - g(\theta^*)], \mathcal{F}_{m'-1}] = 0$; and the third inequality uses the result of Step G.1.

*Step* G.3. During this step, we derive a bound on a vector norm after a single update:

$$
\begin{aligned}
\mathbb{E}[||\theta_t - \theta^*||^2|\mathcal{F}_{m'-1}] &= \mathbb{E}[||\theta_{t-1} - \theta^* + \alpha v_t||^2|\mathcal{F}_{m'-1}] \\
&= \mathbb{E}[||\theta_{t-1} - \theta^*||^2 + 2\alpha(\theta_{t-1} - \theta^*)^T v_t + \alpha^2||v_t||^2|\mathcal{F}_{m'-1}] \\
&\leq \mathbb{E}[||\theta_{t-1} - \theta^*||^2 + 2\alpha(\theta_{t-1} - \theta^*)^T \bar{g}(\theta_{t-1}) + 2\alpha(\theta_{t-1} - \theta^*)^T \eta_{m'} \\
&\quad 4\alpha^2 f_d(\theta_{t-1}) + 4\alpha^2 f_d(\tilde{\theta}) + 2\alpha^2||\eta_{m'}||^2|\mathcal{F}_{m'-1}] \\
&= \mathbb{E}[||\theta_{t-1} - \theta^*||^2 - 2\alpha f_d(\theta_{t-1}) + 2\alpha(\theta_{t-1} - \theta^*)^T \eta_{m'} \\
&\quad + 4\alpha^2 f_d(\theta_{t-1}) + 4\alpha^2 f_d(\tilde{\theta}) + 2\alpha^2||\eta_{m'}||^2|\mathcal{F}_{m'-1}],
\end{aligned}
$$

where the first inequality uses

$$
\begin{aligned}
\mathbb{E}[2\alpha(\theta_{t-1} - \theta^*)^T v_t|\mathcal{F}_{m'-1}] &= \mathbb{E}[2\alpha(\theta_{t-1} - \theta^*)^T (g_t(\theta_{t-1}) - g_t(\tilde{\theta}) + \bar{g}(\tilde{\theta}) + \eta_{m'})|\mathcal{F}_{m'-1}] \\
&= \mathbb{E}[2\alpha(\theta_{t-1} - \theta^*)^T (\bar{g}(\theta_{t-1}) - \bar{g}(\tilde{\theta}) + \bar{g}(\tilde{\theta}) + \eta_{m'})|\mathcal{F}_{m'-1}],
\end{aligned}
$$

and the last equality uses (5). Rearranging terms we obtain:

$$
\begin{aligned}
&\mathbb{E}[||\theta_t - \theta^*||^2 + 2\alpha f_d(\theta_{t-1}) - 4\alpha^2 f_d(\theta_{t-1})|\mathcal{F}_{m'-1}] \\
&\leq \mathbb{E}[||\theta_{t-1} - \theta^*||^2 + 4\alpha^2 f_d(\tilde{\theta}) + 2\alpha(\theta_{t-1} - \theta^*)^T \eta_{m'} + 2\alpha^2||\eta_{m'}||^2|\mathcal{F}_{m'-1}] \\
&\leq \mathbb{E}[||\theta_{t-1} - \theta^*||^2 + 4\alpha^2 f_d(\tilde{\theta}) + 2\alpha||\theta_{t-1} - \theta^*|| \cdot ||\eta_{m'}|| + 2\alpha^2||\eta_{m'}||^2|\mathcal{F}_{m'-1}]. \\
&\leq \mathbb{E}[||\theta_{t-1} - \theta^*||^2 + 4\alpha^2 f_d(\tilde{\theta}) + 2\alpha(\frac{\lambda_A}{2}||\theta_{t-1} - \theta^*||^2 + \frac{1}{2\lambda_A}||\eta_{m'}||^2) + 2\alpha^2||\eta_{m'}||^2|\mathcal{F}_{m'-1}].
\end{aligned}
$$

*Step* G.4. Now derive a bound on epoch update. We use similar logic as during the proof of Theorem 5.1. Since the error term doesn't change over the epoch, summing over the epoch we have:

$$\mathbb{E}[||\theta_{m'} - \theta^*||^2 + 2\alpha M f_d(\tilde{\theta}_{m'}) - 4\alpha^2 M f_d(\tilde{\theta}_{m'})|\mathcal{F}_{m'-1}] \leq$$

$$||\theta_0 - \theta^*||^2 + 4\alpha^2 M f_d(\tilde{\theta}_{m'-1}) + \mathbb{E}[2\alpha \sum_{t=1}^{M} (\frac{\lambda_A}{2}||\theta_{t-1} - \theta^*||^2 + \frac{1}{2\lambda_A}||\eta_{m'}||^2) + 2\alpha^2 M||\eta_{m'}||^2|\mathcal{F}_{m'-1}] =$$

$$||\tilde{\theta}_{m'-1} - \theta^*||^2 + 4\alpha^2 M f_d(\tilde{\theta}_{m'-1}) + \mathbb{E}[2\alpha M(\frac{\lambda_A}{2}||\tilde{\theta}_{m'} - \theta^*||^2 + \frac{1}{2\lambda_A}||\eta_{m'}||^2) + 2\alpha^2 M||\eta_{m'}||^2|\mathcal{F}_{m'-1}] \leq$$

$$\frac{1}{\lambda_A} f_d(\tilde{\theta}_{m'-1}) + 4\alpha^2 M f_d(\tilde{\theta}_{m'-1}) + \mathbb{E}[2\alpha M\left(\frac{1}{2}f_d(\tilde{\theta}_{m'}) + \frac{1}{2\lambda_A}||\eta_{m'}||^2\right) + 2\alpha^2 M||\eta_{m'}||^2|\mathcal{F}_{m'-1}].$$

Rearranging terms, dropping $\mathbb{E}||\theta_{m'} - \theta^*||^2$ and dividing by $2\alpha M$ we further obtain:

$$\left(\frac{1}{2} - 2\alpha\right)\mathbb{E}[f_d(\tilde{\theta}_{m'})|\mathcal{F}_{m'-1}] \leq$$

$$\left(\frac{1}{2\alpha M\lambda_A} + 2\alpha\right)f_d(\tilde{\theta}_{m'-1}) + \left(\frac{1}{2\lambda_a} + \alpha\right)\mathbb{E}[||\eta_{m'}||^2|\mathcal{F}_{m'-1}]$$

Dividing both sides of this equation to $0.5 - 2\alpha$ we have the epoch convergence:

$$\mathbb{E}[f_d(\tilde{\theta}_{m'})|\mathcal{F}_{m'-1}] \leq \left(\frac{1}{\lambda_A 2\alpha M(0.5 - 2\alpha)} + \frac{2\alpha}{0.5 - 2\alpha}\right)f_d(\tilde{\theta}_{m'-1}) + \left(\frac{1}{2\lambda_a(0.5 - 2\alpha)} + \frac{\alpha}{0.5 - 2\alpha}\right)\mathbb{E}[||\eta_{m'}||^2|\mathcal{F}_{m'-1}]. \tag{8}$$

To achieve convergence, we need to guarantee the linear convergence of the first and second terms in the sum separately. The first term is dependent on inner loop updates; its convergence is analyzed in Theorem 5.1. Here we show how to achieve a similar geometric convergence rate of the second term. Since the error term has 0 mean and we are in a finite sample case with replacement, the expected squared norm can be bounded by:

$$\mathbb{E}||\eta_{m'}||^2 \leq \frac{N - n_{m'}}{N n_{m'}}S^2 \leq \left(1 - \frac{n_{m'}}{N}\right)\frac{S^2}{n_{m'}} \leq \frac{S^2}{n_{m'}},$$

where $S^2$ is a bound on the update vector norm variance. If we want the error to be bounded by $c\rho^m$, we need the estimation batch size $n_{m'}$ to satisfy the condition:

$$n_{m'} \geq \frac{S^2}{c\rho^{m'}}.$$

until growing batch size reaches sample size. Satisfying this condition, guarantees that the second term has geometric convergence:

$$\left(\frac{1}{2\lambda_a(0.5 - 2\alpha)} + \frac{\alpha}{0.5 - 2\alpha}\right)\mathbb{E}||\eta_{m'}||^2 \leq \left(\frac{1}{2\lambda_a(0.5 - 2\alpha)} + \frac{\alpha}{0.5 - 2\alpha}\right)c\rho^m.$$

It remains to derive a bound $S^2$ for the update vector norm sample variance:

$$\frac{1}{N-1}\sum_{s,s'}||g_{s,s'}(\theta)||^2 - ||\bar{g}(\theta)||^2 \leq$$

$$\frac{N}{N-1}\frac{1}{N}\sum_{s,s'}||g_{s,s'}(\theta)||^2 = \frac{N}{N-1}\frac{1}{N}\sum_{s,s'}||g_{s,s'}(\theta) - g_{s,s'}(\theta^*) + g_{s,s'}(\theta^*)||^2 \leq$$

$$\frac{N}{N-1}\frac{1}{N}\sum_{s,s'}2||g_{s,s'}(\theta) - g_{s,s'}(\theta^*)||^2 + 2||g_{s,s'}(\theta^*)||^2 =$$

$$\frac{N}{N-1}(2w(\theta) + 2\sigma^2) \leq \frac{N}{N-1}(4f(\theta) + 2\sigma^2) = S^2,$$

where $\sigma^2$ is the variance of the updates in the optimal point similar to Bhandari et al. (2018).

Alternatively, we might derive a bound $S^2$ in terms of quantities known during the algorithm execution:

$$\frac{1}{N-1}\sum_{s,s'}||g_{s,s'}(\theta)||^2 - ||\bar{g}(\theta)||^2 \leq$$

$$\frac{N}{N-1}\frac{1}{N}\sum_{s,s'}||g_{s,s'}(\theta)||^2 = \frac{N}{N-1}\frac{1}{N}\sum_{s,s'}||(r(s,s') + \gamma\phi(s')^T\theta - \phi(s)^T\theta)\phi(s)||^2 \leq$$

$$\frac{N}{N-1}\frac{1}{N}\sum_{s,s'}2||r\phi(s)||^2 + 4||\gamma\phi(s')^T\theta\phi(s)||^2 + 4||\phi(s)^T\theta\phi(s)||^2 \leq$$

$$\frac{N}{N-1}(2|r_{max}|^2 + 4\gamma^2||\theta||^2 + 4||\theta||^2) = \frac{N}{N-1}(2|r_{max}|^2 + 8||\theta||^2) = S^2.$$

Having the convergence of the both terms of 8, we proceed by expanding the equation for earlier epochs (denoting bracket terms as $\rho$ and $\rho'$):

$$\mathbb{E}[f_d(\tilde{\theta}_{m'})|\mathcal{F}_{m'-1}] \leq \rho f_d(\tilde{\theta}_{m'-1}) + \rho'\mathbb{E}[||\eta_{m'}||^2|\mathcal{F}_{m'-1}] \implies$$
$$\mathbb{E}[f_d(\tilde{\theta}_{m'})|\mathcal{F}_{m-2}] \leq \rho^2 f_d(\tilde{\theta}_{m-2}) + \rho'(\mathbb{E}[||\eta_{m'}||^2|\mathcal{F}_{m-2}] + \rho\mathbb{E}[||\eta_{m'}||^2|\mathcal{F}_{m'-1}]) \implies$$
$$\mathbb{E}[f_d(\tilde{\theta}_{m'})] \leq \rho^m f_d(\tilde{\theta}_0) + \rho'(\sum_{i=1}^{m}\rho^i\mathbb{E}[||\eta_i||^2|\mathcal{F}_i])$$

Now, assuming that estimation batch sizes are large enough that all error terms are bounded by $c\rho^m$:

$$\mathbb{E}[f_d(\tilde{\theta}_{m'})|\mathcal{F}_{m'-1}] \leq \rho^m f_d(\tilde{\theta}_0) + \rho'(\sum_{i=1}^{m}\rho^i c\rho^m) \leq \rho^m f_d(\tilde{\theta}_0) + \rho^m\frac{c\rho'}{1-\rho}.$$

Denoting $\frac{c\rho'}{1-\rho}$ as $C$ we have the claimed result.

## H    Proof of Theorem 5.4

---

**Algorithm 3** TD-SVRG for the i.i.d. sampling case

---

**Parameters** update batch size $M$ and learning rate $\alpha$.
**Initialize** $\tilde{\theta}_0$.
**for** $m' = 1, 2, ..., m$ **do**
    $\tilde{\theta} = \tilde{\theta}_{m'-1}$,
    choose estimation batch size $n_{m'}$,
    sample batch $\mathcal{D}^{m'}$ of size $n_{m'}$,
    compute $g_{m'}(\tilde{\theta}) = \frac{1}{n_{m'}} \sum_{s,s' \in \mathcal{D}^{m'}} g_{s,s'}(\tilde{\theta})$,
    where $g_{s,s'}(\tilde{\theta}) = (r(s, s') + \gamma \phi(s')^T \tilde{\theta} - \phi(s)^T \tilde{\theta}) \phi(s_t)$.
    $\theta_0 = \tilde{\theta}$.
    **for** $t = 1$ **to** $M$ **do**
        Sample $s, s'$ from $\mathcal{D}$.
        Compute $v_t = g_{s,s'}(\theta_{t-1}) - g_{s,s'}(\tilde{\theta}) + g_{m'}(\tilde{\theta})$.
        Update parameters $\theta_t = \theta_{t-1} + \alpha v_t$.
    **end for**
    Set $\tilde{\theta}_{m'} = \theta_{t'}$ for randomly chosen $t' \in (0, \ldots, M-1)$.
**end for**

---

The proof is very similar to 8, the only difference is that now we derive an expectation with respect to an MDP instead of a finite sample dataset.

*Step* H.1. During the first step we use the bound obtained during the proof of Theorem 5.1:

$$
\begin{aligned}
w(\theta) &= (\theta - \theta^*)^T \mathbb{E}[(\gamma \phi(s') - \phi(s))\phi(s)^T \phi(s)(\gamma \phi(s') - \phi(s))^T](\theta - \theta^*) \\
&= (\theta - \theta^*)^T \Big[ \sum_{s,s'} \mu_\pi(s) P(s, s')(\gamma \phi(s') - \phi(s))\phi^T(s)\phi(s)(\gamma \phi(s') - \phi(s))^T \Big](\theta - \theta^*) \\
&\leq (\theta - \theta^*)^T \Big[ \sum_{s,s'} \mu_\pi(s) P(s, s')(\gamma \phi(s') - \phi(s))(\gamma \phi(s') - \phi(s))^T \Big](\theta - \theta^*) \\
&= (\theta - \theta^*)^T \Big[ \sum_{s,s'} \mu_\pi(s) P(s, s')(\gamma^2 \phi(s')\phi(s')^T - \gamma \phi(s')\phi(s)^T) \Big](\theta - \theta^*) + f_e(\theta) \\
&= (\theta - \theta^*)^T \sum_{s,s'} \mu_\pi(s) P(s, s')(\gamma^2 \phi(s)\phi(s)^T - \gamma \phi(s)\phi(s')^T) \Big](\theta - \theta^*) + f_e(\theta) \\
&\leq 2 f_e(\theta),
\end{aligned}
\tag{9}
$$

where the first inequality uses Assumption 2.2, the third equality uses the fact that $\mu_\pi$ is a stationary distribution of $P$ ($\sum_{s'} \gamma^2 \mu_\pi(s) P(s, s')\phi(s')\phi(s')^T = \sum_{s'} \gamma^2 \mu_\pi(s')\phi(s')\phi(s')^T = \sum_s \mu_\pi(s)\gamma^2 \phi(s)\phi(s)^T$). The last inequality uses the fact that $\gamma < 1$.

*Step* H.2. During this step we derive a bound on the squared norm of a single update $\mathbb{E}[||v_t||^2]$, which is performed during time step $t$ of epoch $m$. Since we are aiming to derive epoch to epoch convergence bound, we will be taking expectation conditioned on all history previous to epoch $m$, which we denote as $\mathcal{F}_{m'-1}$. Similarly with Appendix G.1 we assume that mean path update in the end of the previous epoch was computed inexactly and has estimation error: $\bar{g}(\tilde{\theta}_{m'-1}) + \eta_{m'}$. Thus the single update vector becomes (for simplicity we denote $\tilde{\theta}_{m'-1}$ as $\tilde{\theta}$):

$$
v_t = g_t(\theta_{t-1}) - g_t(\tilde{\theta}) + \bar{g}(\tilde{\theta}) + \eta_{m'}.
$$

The norm of this vector is bounded by:

$$\mathbb{E}[||v_t||^2|\mathcal{F}_{m'-1}] = \mathbb{E}[||g_t(\theta_{t-1}) - g_t(\tilde{\theta}) + \bar{g}(\tilde{\theta}) + \eta_{m'}||^2|\mathcal{F}_{m'-1}]$$

$$= \mathbb{E}[||(g_t(\theta_{t-1}) - \bar{g}(\theta^*)) + (\bar{g}(\theta^*) - g_t(\tilde{\theta}) + \bar{g}(\tilde{\theta}) + \eta_{m'})||^2|\mathcal{F}_{m'-1}]$$

$$\leq 2\mathbb{E}[||(g_t(\theta_{t-1}) - g_t(\theta^*))||^2|\mathcal{F}_{m'-1}]+$$

$$2\mathbb{E}[||g_t(\tilde{\theta}) - g_t(\theta^*) - (\bar{g}(\tilde{\theta}) - \bar{g}(\theta^*)) - \eta_{m'}||^2|\mathcal{F}_{m'-1}]$$

$$= 2\mathbb{E}[||(g_t(\theta_{t-1}) - g_t(\theta^*))||^2|\mathcal{F}_{m'-1}]+$$

$$2\mathbb{E}[||g_t(\tilde{\theta}) - g_t(\theta^*) - \mathbb{E}[g_t(\tilde{\theta}) - g_t(\theta^*)] - \eta_{m'}||^2|\mathcal{F}_{m'-1}]$$

$$= 2\mathbb{E}[||(g_t(\theta_{t-1}) - g_t(\theta^*))||^2|\mathcal{F}_{m'-1}]+$$

$$2\mathbb{E}[||g_t(\tilde{\theta}) - g_t(\theta^*) - \mathbb{E}[g_t(\tilde{\theta}) - g_t(\theta^*)]||^2|\mathcal{F}_{m'-1}]$$

$$- \mathbb{E}[\langle g_t(\tilde{\theta}) - g_t(\theta^*) - \mathbb{E}[g_t(\tilde{\theta}) - g_t(\theta^*)], \eta_{m'}\rangle|\mathcal{F}_{m'-1}] + 2\mathbb{E}[||\eta_{m'}||^2|\mathcal{F}_{m'-1}]$$

$$\leq 2\mathbb{E}[||(g_t(\theta_{t-1}) - g_t(\theta^*))||^2|\mathcal{F}_{m'-1}]+$$

$$2\mathbb{E}[||g_t(\tilde{\theta}) - g_t(\theta^*)||^2|\mathcal{F}_{m'-1}] + 2\mathbb{E}[||\eta_{m'}||^2|\mathcal{F}_{m'-1}]$$

$$= 2\mathbb{E}[w(\theta_{t-1}) + 2w(\tilde{\theta}) + 2||\eta_{m'}||^2|\mathcal{F}_{m'-1}]$$

$$\leq \mathbb{E}[4f_e(\theta_{t-1}) + 4f_e(\tilde{\theta}) + 2||\eta_{m'}||^2|\mathcal{F}_{m'-1}],$$

where the first inequality uses $\mathbb{E}||A+B||^2 \leq 2\mathbb{E}||A||^2 + 2\mathbb{E}||B||^2$, the second inequality uses $\mathbb{E}||A - \mathbb{E}[A]||^2 \leq \mathbb{E}||A||^2$ and the fact $\mathbb{E}[\eta_{m'}|g(\tilde{\theta}) - g(\theta^*) - \mathbb{E}_{s,s'}[g(\tilde{\theta}) - g(\theta^*)], \mathcal{F}_{m'-1}] = 0$ ; and the third inequality uses the result of Step H.1.

*Step* H.3. We obtain a bound on a vector norm after a single update during iteration $t$ of epoch $m$:

$$\mathbb{E}[||\theta_t - \theta^*||^2|\mathcal{F}_{m'-1}] = \mathbb{E}[||\theta_{t-1} - \theta^* + \alpha v_t||^2|\mathcal{F}_{m'-1}]$$

$$= \mathbb{E}[||\theta_{t-1} - \theta^*||^2 + 2\alpha(\theta_{t-1} - \theta^*)^T v_t + \alpha^2||v_t||^2|\mathcal{F}_{m'-1}]$$

$$= \mathbb{E}[||\theta_{t-1} - \theta^*||^2 + 2\alpha(\theta_{t-1} - \theta^*)^T \bar{g}(\theta_{t-1}) + 2\alpha(\theta_{t-1} - \theta^*)^T \eta_{m'}$$

$$+ 4\alpha^2 f_e(\theta_{t-1}) + 4\alpha^2 f_e(\tilde{\theta}) + 2\alpha^2||\eta_{m'}||^2|\mathcal{F}_{m'-1}].$$

Applying an argument similar to 5 and rearranging terms we obtain:

$$\mathbb{E}[||\theta_t - \theta^*||^2 + 2\alpha f_e(\theta_{t-1}) - 4\alpha^2 f_e(\theta_{t-1})|\mathcal{F}_{m'-1}]$$

$$\leq \mathbb{E}[||\theta_{t-1} - \theta^*||^2 + 4\alpha^2 f_e(\tilde{\theta}) - 2\alpha(\theta_{t-1} - \theta^*)^T \eta + 2\alpha^2||\eta_{m'}||^2|\mathcal{F}_{m'-1}]$$

$$\leq \mathbb{E}[||\theta_{t-1} - \theta^*||^2 + 4\alpha^2 f_e(\tilde{\theta}) + 2\alpha||\theta_{t-1} - \theta^*|| \cdot ||\eta|| + 2\alpha^2||\eta_{m'}||^2|\mathcal{F}_{m'-1}].$$

$$\leq \mathbb{E}[||\theta_{t-1} - \theta^*||^2 + 4\alpha^2 f_e(\tilde{\theta}) + 2\alpha(\frac{\lambda_A}{2}||\theta_{t-1} - \theta^*||^2 + \frac{1}{2\lambda_A}||\eta_{m'}||^2) + 2\alpha^2||\eta_{m'}||^2|\mathcal{F}_{m'-1}].$$

*Step* H.4. Now derive a bound on epoch update. We use the similar logic as during the proof of Theorem 5.1. Since the error term doesn't change over the epoch, summing over the epoch we have:

$$\mathbb{E}[||\theta_{m'} - \theta^*||^2 + 2\alpha M f_e(\tilde{\theta}_{m'}) - 4\alpha^2 M f_e(\tilde{\theta}_{m'})|\mathcal{F}_{m'-1}] \leq$$

$$||\theta_0 - \theta^*||^2 + 4\alpha^2 M f_e(\tilde{\theta}_{m'-1}) + \mathbb{E}[2\alpha \sum_{t=1}^{M}(\frac{\lambda_A}{2}||\theta_{t-1} - \theta^*||^2 + \frac{1}{2\lambda_A}||\eta_{m'}||^2) + 2\alpha^2 M||\eta_{m'}||^2|\mathcal{F}_{m'-1}] =$$

$$||\tilde{\theta}_{m'-1} - \theta^*||^2 + 4\alpha^2 M f_e(\tilde{\theta}_{m'-1}) + \mathbb{E}[2\alpha M(\frac{\lambda_A}{2}||\tilde{\theta}_{m'} - \theta^*||^2 + \frac{1}{2\lambda_A}||\eta_{m'}||^2) + 2\alpha^2 M||\eta_{m'}||^2|\mathcal{F}_{m'-1}] \leq$$

$$\frac{1}{\lambda_a} f_e(\tilde{\theta}_{m'-1}) + 4\alpha^2 M f_e(\tilde{\theta}_{m'-1}) + \mathbb{E}[2\alpha M\left(\frac{1}{2}f_e(\tilde{\theta}_{m'}) + \frac{1}{2\lambda_A}||\eta_{m'}||^2\right) + 2\alpha^2 M||\eta_{m'}||^2|\mathcal{F}_{m'-1}]$$

Rearranging terms, dropping $\mathbb{E}||\theta_{m'} - \theta^*||^2$ and dividing by $2\alpha M$ we further obtain:

$$\left(\frac{1}{2} - 2\alpha\right)\mathbb{E}[f_e(\tilde{\theta}_{m'})|\mathcal{F}_{m'-1}] \leq$$
$$\left(\frac{1}{2\alpha M\lambda_A} + 2\alpha\right)f_e(\tilde{\theta}_{m'-1}) + \left(\frac{1}{2\lambda_a} + \alpha\right)\mathbb{E}[||\eta_{m'}||^2|\mathcal{F}_{m'-1}]$$

Dividing both sides of this equation to $0.5 - 2\alpha$ we have the epoch convergence:

$$\mathbb{E}[f_e(\tilde{\theta}_{m'})|\mathcal{F}_{m'-1}] \leq \left(\frac{1}{\lambda_A 2\alpha M(0.5 - 2\alpha)} + \frac{2\alpha}{0.5 - 2\alpha}\right)f_e(\tilde{\theta}_{m'-1}) +$$
$$\left(\frac{1}{2\lambda_a(0.5 - 2\alpha)} + \frac{\alpha}{0.5 - 2\alpha}\right)\mathbb{E}[||\eta_{m'}||^2|\mathcal{F}_{m'-1}].$$

Similarly to Appendix D, convergence for the first term might be obtained by setting the learning rate to $\alpha = 1/16$ and the update batch size to $M = 32/\lambda_A$. To guarantee convergence of the second term, we need to bound $\mathbb{E}||\eta_{m'}||^2$. In the infinite population with replacement case, the norm of the error vector is bounded by:

$$\mathbb{E}||\eta_{m'}||^2 \leq \frac{S^2}{n_{m'}},$$

where $S^2$ is a bound update vector norm variance. If we want the error to be bounded by $c\rho^m$, we need the estimation batch size $n_{m'}$ to satisfy the condition:

$$n_{m'} \geq \frac{S^2}{c\rho^m}.$$

Satisfying this condition guarantees that the second term has geometric convergence:

$$\left(\frac{1}{2\lambda_a(0.5 - 2\alpha)} + \frac{\alpha}{0.5 - 2\alpha}\right)\mathbb{E}||\eta_{m'}||^2 \leq \left(\frac{1}{2\lambda_a(0.5 - 2\alpha)} + \frac{\alpha}{0.5 - 2\alpha}\right)c\rho^m.$$

Similarly to Appendix G.1, the bound on sample variance $S^2$ can be derived as follows:

$$\sum_{s,s'}\mu_\pi(s)P(s,s')||g_{s,s'}(\theta)||^2 - ||\bar{g}(\theta)||^2 \leq$$
$$\sum_{s,s'}\mu_\pi(s)P(s,s')||g_{s,s'}(\theta) - g_{s,s'}(\theta^*) + g_{s,s'}(\theta^*)||^2 \leq$$
$$\sum_{s,s'}\mu_\pi(s)P(s,s')2||g_{s,s'}(\theta) - g_{s,s'}(\theta^*)||^2 + 2||g_{s,s'}(\theta^*)||^2 =$$
$$2w(\theta) + 2\sigma^2 \leq 4f(\theta) + 2\sigma^2 = S^2,$$

where $\sigma^2$ is the variance of the updates in the optimal point similar to Bhandari et al. (2018).

An alternative bound on $S^2$ with known quantities for practical implementation:

$$\sum_{s,s'}\mu_\pi(s)P(s,s')||g_{s,s'}(\theta)||^2 - ||\bar{g}(\theta)||^2 \leq$$
$$\sum_{s,s'}\mu_\pi(s)P(s,s')||g_{s,s'}(\theta)||^2 = \sum_{s,s'}\mu_\pi(s)P(s,s')(||(r(s,s') + \gamma\phi(s')^T\theta - \phi(s)^T\theta)\phi(s)||^2) \leq$$
$$\sum_{s,s'}\mu_\pi(s)P(s,s')(2||r\phi(s)||^2 + 4||\gamma\phi(s')^T\theta\phi(s)||^2 + 4||\phi(s)^T\theta\phi(s)||^2) \leq$$
$$(2|r_{max}|^2 + 4\gamma^2||\theta||^2 + 4||\theta||^2) = (2|r_{max}|^2 + 8||\theta||^2) = S^2.$$

Setting hyperparameters to obtained values will results in final computational complexity of $\mathcal{O}(\frac{1}{\epsilon \lambda_A} \log(\epsilon^{-1}))$.

# I Proof of Theorem 5.6

---
**Algorithm 4** TD-SVRG for the Markovian sampling case

---
**Parameters** update batch size $M$ and learning rate $\alpha$ and projection radius $R$.
**Initialize** $\tilde{\theta}_0$.
**for** $m = 1, 2, ..., m$ **do**
    $\tilde{\theta} = \tilde{\theta}_{m'-1}$,
    choose estimation batch size $n_{m'}$,
    sample trajectory $\mathcal{D}^{m'}$ of length $n_{m'}$,
    compute $g_{m'}(\tilde{\theta}) = \frac{1}{n_{m'}} \sum_{s,s' \in \mathcal{D}^{m'}} g_{s,s'}(\tilde{\theta})$,
    where $g_{s,s'}(\tilde{\theta}) = (r(s,s') + \gamma\phi(s')^T\tilde{\theta} - \phi(s)^T\tilde{\theta})\phi(s_t)$.
    $\theta_0 = \tilde{\theta}$.
    **for** $t = 1$ **to** $M$ **do**
        Sample $s, s'$ from $\mathcal{D}$.
        Compute $v_t = g_{s,s'}(\theta_{t-1}) - g_{s,s'}(\tilde{\theta}) + g_{m'}(\tilde{\theta})$.
        Update parameters $\theta_t = \Pi_R(\theta_{t-1} + \alpha v_t)$.
    **end for**
    Set $\tilde{\theta}_{m'} = \theta_{t'}$ for randomly chosen $t' \in (0, \ldots, M-1)$.
**end for**

---

In this section we show the convergence of the Algorithm 4, which might be applied in the Markovian sampling case. In this case, we cannot simply apply Lemma 4.1; due to high estimation bias the bounds on $f_e(\theta)$ and $w(\theta)$ will not be derived based on the current value of $\theta$, but based on global constraints on the updates guaranteed by applying projection.

First, we analyse a single iteration on step $t$ of epoch $m$, during which we apply the update vector $v_t = g_t(\theta) - g_t(\tilde{\theta}) + g_{m'}(\tilde{\theta})$. The update takes the form:

$$
\begin{aligned}
\mathbb{E}||\theta_t - \theta^*||_2^2 = \mathbb{E}||\Pi_R(\theta_{t-1} + \alpha v_t) - \Pi_R(\theta^*)||_2^2 \leq \mathbb{E}||\theta_{t-1} - \theta^* + (-\alpha v_t)||_2^2 = \\
||\theta_{t-1} - \theta^*||_2^2 + 2\alpha(\theta_{t-1} - \theta^*)^T \mathbb{E}[v_t] + \alpha^2 E||v_t||_2^2 = \\
||\theta_{t-1} - \theta^*||_2^2 + 2\alpha(\theta_{t-1} - \theta^*)^T (\mathbb{E}[g_t(\theta_{t-1})] - \mathbb{E}[g_t(\tilde{\theta})] + g_{m'}(\tilde{\theta})) + \\
\alpha^2 E||v_t||_2^2,
\end{aligned}
\tag{10}
$$

where the expectation is taken with respect to $s, s'$ sampled during iteration $t$. Recall that under Markovian sampling, $\mathbb{E}[g_t(\theta_{t-1})] \neq \bar{g}(\theta_{t-1})$ and that for the expectation of the estimated mean-path update we have $\mathbb{E}[g_{m'}(\tilde{\theta})|s_{m'-1}] \neq \bar{g}(\tilde{\theta})$, where $s_{m'-1}$ is the last state of epoch $m-1$. To tackle this issue, we follow the approach introduced in Bhandari et al. (2018) and Xu et al. (2020), and rewrite the expectation as a sum of mean-path updates and error terms. Similar to Bhandari et al. (2018), we denote the error term on a single update as $\zeta$:

$$
\zeta_t(\theta) = (\theta - \theta^*)^T (g_t(\theta) - \bar{g}(\theta)).
$$

For an error term on the trajectory, we follow Xu et al. (2020) and denote it as $\xi$:

$$
\xi_{m'}(\theta, \tilde{\theta}) = (\theta - \theta^*)^T (g_{m'}(\tilde{\theta}) - \bar{g}(\theta)).
$$

Applying this notation, (10) can be rewritten as:

$$
\begin{aligned}
E||\theta_t - \theta^*||_2^2 \leq & ||\theta_{t-1} - \theta^*||_2^2 + \\
& 2\alpha(\theta_{t-1} - \theta^*)^T (\mathbb{E}[g_{t-1}(\theta_{t-1})] - \mathbb{E}[g_t(\tilde{\theta})] + g_{m'}(\tilde{\theta})) + \alpha^2 E||v_t||_2^2 = \\
& ||\theta_{t-1} - \theta^*||_2^2 + 2\alpha[(E[\zeta_t(\theta_{t-1})] + (\theta_{t-1} - \theta^*)^T \bar{g}(\theta_{t-1})) - \\
& (E[\zeta_t(\tilde{\theta})] - (\theta_{t-1} - \theta^*)^T \bar{g}(\tilde{\theta})) + \\
& (E[\xi(\theta_{t-1}, \tilde{\theta})] - (\theta_{t-1} - \theta^*)^T \bar{g}(\tilde{\theta}))] + \alpha^2 E||v_t||_2^2.
\end{aligned}
\tag{11}
$$

The error terms can be bounded by slightly modified lemmas from the original papers. For $\zeta(\theta)$, we apply a bound from Bhandari et al. (2018), Lemma 11:

$$|E[\zeta_t(\theta)]| \leq G^2(4 + 6\tau^{mix}(\alpha))\alpha. \tag{12}$$

In the original lemma, a bound on $E[\zeta_t(\theta)]$ is stated, however, in the proof a bound on absolute value of the expectation is also derived.

For the mean-path estimation error term, we use a modified version of Lemma 1 Xu et al. (2020). The proof of this lemma in the original paper starts by applying the inequality

$$a^T b \leq \frac{k}{2}||a||^2 + \frac{1}{2k}||b||^2$$

to the expression $(\theta_{t-1} - \theta^*)^T(g_{m'}(\tilde{\theta}) - \bar{g}(\theta))$, with $k = \lambda_A/2$ (using the notation in Xu et al. (2020)). For the purposes of our proof we use $k = \lambda_A$. Thus, we will have the expression:

$$
\begin{aligned}
\mathbb{E}[\xi_{m'}(\theta_{t-1}, \tilde{\theta})] \leq &\frac{\lambda_A}{2}\mathbb{E}[||\theta_{t-1} - \theta^*||_2^2|s_{m'-1}] + \frac{4(1 + (m-1)\rho)}{\lambda_A(1-\rho)n_{m'}}[4R^2 + r_{\max}^2] = \\
&\frac{\lambda_A}{2}\mathbb{E}[||\theta_{t-1} - \theta^*||_2^2|s_{m'-1}] + \frac{C_2}{\lambda_A n_{m'}}.
\end{aligned}
\tag{13}
$$

Also, note, that the term $E||v_t||_2^2$ might be bounded as $E||v_t||_2^2 \leq 18R^2$. Plugging these bounds into (11) we obtain:

$$
\begin{aligned}
E||\theta_t - \theta^*||_2^2 \leq &||\theta_{t-1} - \theta^*||_2^2 - 2\alpha f_e(\theta_{t-1}) + 4\alpha^2 G^2(4 + 6\tau^{mix}(\alpha)) + \\
&2\alpha\left(\frac{\lambda_A}{2}||\theta_{t-1} - \theta^*||_2^2 + \frac{C_2}{\lambda_A n_{m'}}\right) + 18\alpha^2 R^2.
\end{aligned}
$$

Summing the inequality over the epoch and taking expectation with respect to all previous history, we have:

$$
\begin{aligned}
2\alpha M \mathbb{E}[f_e(\tilde{\theta}_s)] \leq &||\tilde{\theta}_{s-1} - \theta^*||_2^2 + 2\alpha M\left(\frac{\lambda_A}{2}||\tilde{\theta}_{s-1} - \theta^*||_2^2 + \frac{C_2}{\lambda_A n_{m'}}\right) + \\
&\alpha^2 M(4G^2(4 + 6\tau^{mix}(\alpha)) + 18R^2).
\end{aligned}
$$

Then we divide both sides by $2\alpha M$ and use $||\tilde{\theta}_{s-1} - \theta^*||_2^2 \leq f_e(\tilde{\theta}_{s-1})/\lambda_A$ to obtain:

$$
\begin{aligned}
E[f_e(\tilde{\theta}_s)] \leq &\left(\frac{1}{2\lambda_A \alpha M} + \frac{1}{2}\right)f_e(\tilde{\theta}_{s-1}) + \frac{C_2}{\lambda_A n_{m'}} + \\
&\alpha(2G^2(4 + 6\tau^{mix}(\alpha)) + 9R^2).
\end{aligned}
$$

We choose $\alpha$ and $M$ such that $\alpha M \lambda_A = 2$. We then apply this inequality to the value of the function $f$ in the first term of the right-hand side recursively, which yields the desired result:

$$E[f_e(\tilde{\theta}_s)] \leq \left(\frac{3}{4}\right)^s f_e(\theta_0) + \frac{8C_2}{\lambda_A n_{m'}} + 4\alpha(2G^2(4 + 6\tau^{mix}(\alpha)) + 9R^2).$$

# J  Additional experiments

## J.1  Comparison of theoretic batchsizes

In this subsection, we compare the values of update batch sizes which are theoretically required to guarantee convergence. We compare batch sizes of three algorithms: TD-SVRG, PDSVRG (Du et al. (2017)) and VRTD (Xu et al. (2020)). Note that PDSVRG and VRTD are algorithms for different settings, but for TD-SVRG the batch size value is the same: $16/\lambda_A$, thus, we compare two algorithms in the same table. We compare the batch sizes required by the algorithm for three MDPs: a first MDP with 50 state, 20 actions and $\gamma = 0.8$, a second MDP with 400 states, 10 actions and $\gamma = 0.95$, and a third MDP with 1000 states, 20 actions and $\gamma = 0.99$, with actions selection probabilities generated from $U[0, 1)$ (similar to the settings used for the experiments in Sections 6 and J.5). Since the batch size is dependent on the smallest eigenvalue of the matrix $A$, which, in turn, is dependent on the dimensionality of the feature vector, we do the comparison for different feature vector sizes: 5, 10, 20 and 40 randomly generated features and 1 constant feature for each state. We generate 10 datasets and environments for each feature size. Our results are summarized in Figure 3 and Tables 3, 2 and 5.

Table 3: Comparison of theoretically suggested batch sizes for an MDP with 50 states, 20 actions and $\gamma = 0.8$. Values in the first row indicate the demensionality of the feature vectors. Values in the other rows: batch size of the corresponding method. Values are averaged over 10 generated datasets and environments.

| Method/Features | 6 | 11 | 21 | 41 |
|---|---|---|---|---|
| TD-SVRG | 2339 | 6808 | 21553 | $4.51 \cdot 10^5$ |
| PD-SVRG | $1.52 \cdot 10^{16}$ | $3.09 \cdot 10^{19}$ | $1.85 \cdot 10^{23}$ | $1.41 \cdot 10^{36}$ |
| VRTD | $3.07 \cdot 10^6$ | $2.13 \cdot 10^7$ | $3.79 \cdot 10^8$ | $165 \cdot 10^{11}$ |

Table 4: Comparison of theoretically suggested batch sizes for an MDP with 400 states, 10 actions and $\gamma = 0.95$. Values in the first row indicate the demensionality of the feature vectors. Values in the other rows: batch size of the corresponding method. Values are averaged over 10 generated datasets and environments.

| Method/Features | 6 | 11 | 21 | 41 |
|---|---|---|---|---|
| TD-SVRG | 3176 | 6942 | 18100 | 54688 |
| PD-SVRG | $1.72 \cdot 10^{16}$ | $3.83 \cdot 10^{18}$ | $3.06 \cdot 10^{21}$ | $5.77 \cdot 10^{24}$ |
| VRTD | $5.41 \cdot 10^6$ | $2.53 \cdot 10^7$ | $1.63 \cdot 10^8$ | $1.58 \cdot 10^9$ |

Table 5: Comparison of theoretically suggested batch sizes for an MDP with 1000 states, 20 actions and $\gamma = 0.99$. Values in the first row indicate the demensionality of the feature vectors. Values in the other rows: batch size of the corresponding method. Values are averaged over 10 generated datasets and environments.

| Method/Features | 6 | 11 | 21 | 41 |
|---|---|---|---|---|
| TD-SVRG | 9206 | 16096 | 32723 | 79401 |
| PD-SVRG | $7.38 \cdot 10^{18}$ | $9.64 \cdot 10^{20}$ | $5.14 \cdot 10^{23}$ | $4.97 \cdot 10^{26}$ |
| VRTD | $4.35 \cdot 10^7$ | $1.34 \cdot 10^8$ | $5.44 \cdot 10^8$ | $1.45 \cdot 10^9$ |

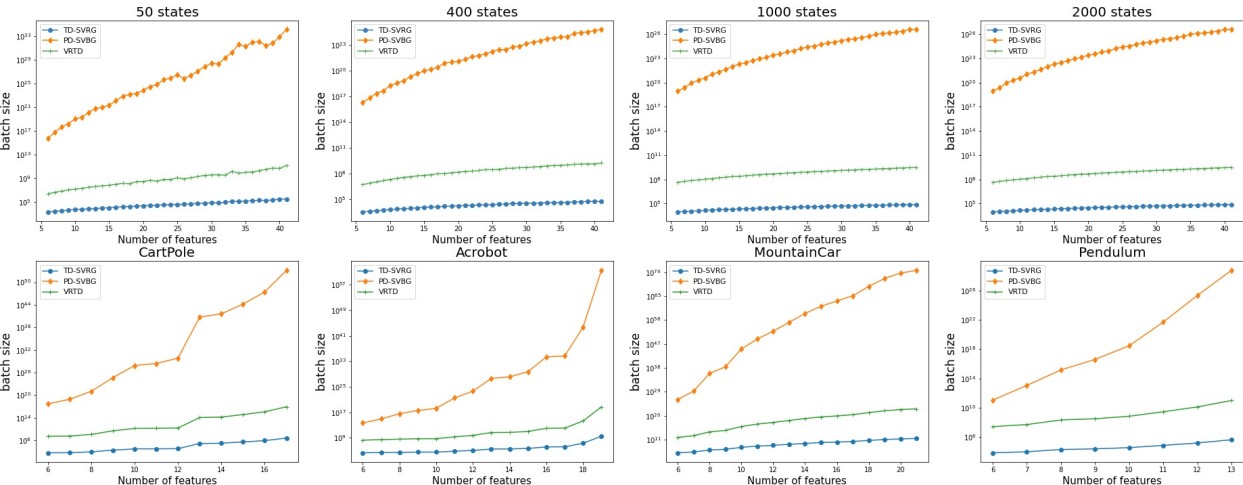

Figure 3: Theoretical batch sizes of different algorithms in **log-scale**, geometrical average over 10 samples. The *x*-axis plots the dimension of the feature vector. *First row:* Batch sizes for random MDP environment (see Sec. 6). Left to right: Figure 1 - 50 states, 20 actions and $\gamma = 0.8$; Figure 2: 400 states, 10 actions and $\gamma = 0.95$, Figure 3: 1000 states, 20 actions and $\gamma = 0.99$; Figure 4: 2000 states, 50 actions and $\gamma = 0.75$. *Second row:* batch sizes for dataset generated from OpenAI gym classic control environments Brockman et al. (2016). Features generated by applying RBF kernels and then removing highly correlated feature vectors one by one (see Sec. 6).

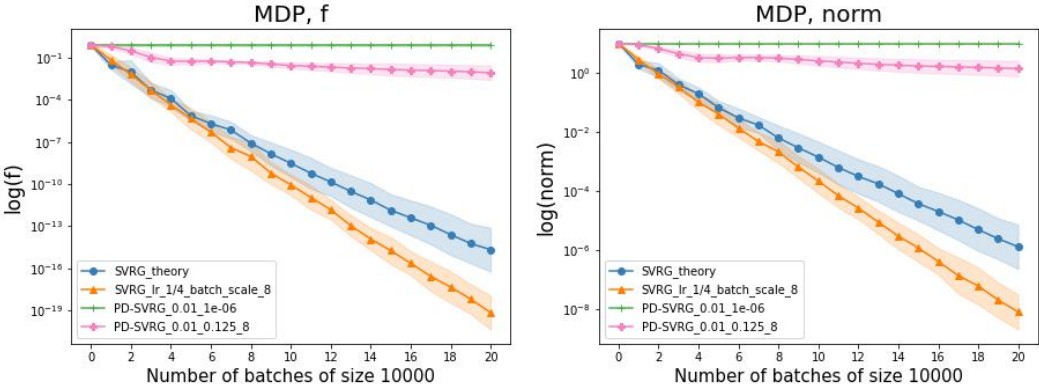

Figure 4: Average performance of TD-SVRG and PD-SVRG algorithms with different parameters: "SVRG_theory" is TD-SVRG algorithm with parameters suggested by theoretical analysis; "SVRG_lr_1/4_batch_scale_8" is a best-performing algorithm from TD-SVRG search grid ($\alpha = 1/4$, $M = 8/\lambda_A$); "PD-SVRG_0.01_1e-6" is a the best perforing algorithm from the first PD-SVRG search grid ($\sigma_\theta = 10^{-6} \frac{1}{L_\rho \kappa(\hat{C})}$, $\sigma_w = 10^{-2} \frac{1}{\lambda_{\max}(C)}$); "PD-SCRG_0.01_0.125_8" is the best performing PS-SVRG algorithms from the second grid search ($\alpha = 1/8$, $M = 8/\lambda_A$, $\sigma_w = 10^{-2} \frac{1}{\lambda_{\max}(C)}$). Rows - performance measurements: $\log(f(\theta))$ and $\log(|\theta - \theta^*|)$.

## J.2 Additional parameter grid search in dataset case

In this set of experiments, we conducted additional grid searches for the TD-SVRG and PD-SVRG algorithms. For TD-SVRG, we executed a grid search on the set of parameters near the theoretically predicted parameters (update batch size $M = 16/\lambda_A$, learning rate $\alpha = 1/8$). For PD-SVRG, we ran searches over two grids: parameters suggested by the authors of the original paper and parameters close to those suggested by our theory. All experiments were conducted on an MDP environment with 400 states, 21

features, 10 actions, and $\gamma = 0.95$, identical to the one described in Section 6 of this paper. For TD-SVRG, we ran a grid search over the parameter batch size $M \in \{8, 12, 16, 24, 32\}/\lambda_A$ and learning rate $\alpha \in \{1/4, 1/6, 1/8, 1/12, 1/16\}$. For the PD-SVRG algorithm, the first grid was formed near the exact values suggested in Du et al. (2017), i.e., primal variables learning rate $\sigma_\theta \in \{10^{-1}, \ldots, 10^{-6}\}/(L_\rho \kappa(\hat{C}))$, dual parameters learning rate $\sigma_w \in \{1, 10^{-1}, 10^{-2}\}/\lambda_{\max}(C)$, and the batch size is twice the dataset size ($M = 2N$). The second grid uses the same learning rate and batch sizes as TD-SVRG, with the dual parameters learning rate $\sigma_w$ being the same as in the previous grid.

The results are illustrated in Figure 4. This figure demonstrates that TD-SVRG converges faster than the PD-SVRG algorithm, which utilizes dual variables.

### J.3 Datasets with DQN features

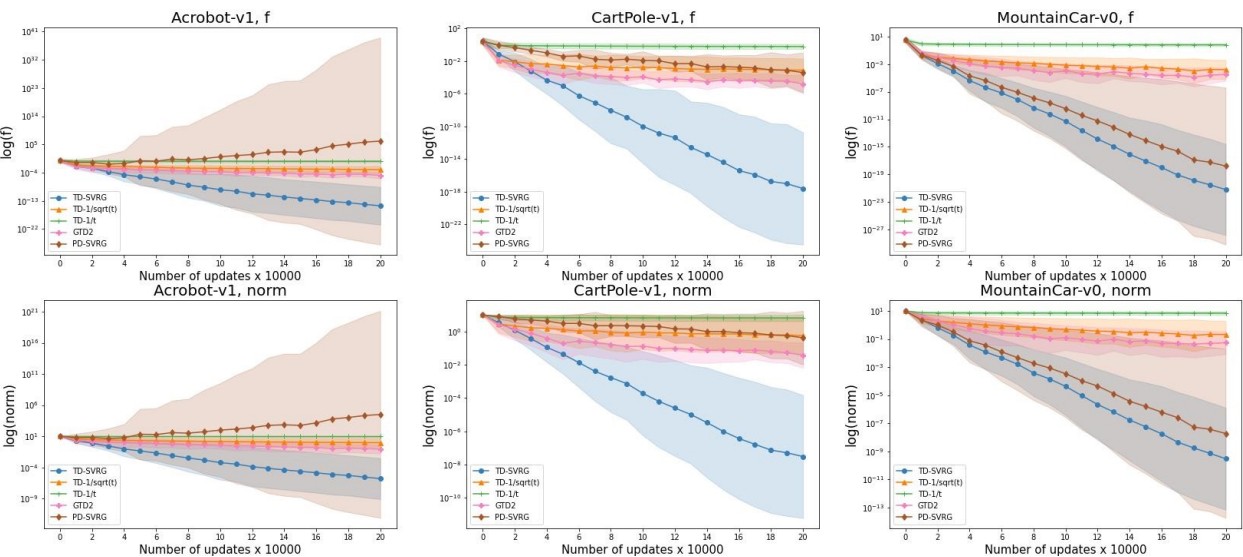

Figure 5: Geometric average performance of different algorithms in the finite sample case with DQN features. Columns - dataset source environments: Acrobot, CartPole and Mountain Car. Rows - performance measurements: $\log(f(\theta))$ and $\log(|\theta - \theta^*|)$.

In this set of experiments, we compare the performance of the same algorithms as in Section 6 on datasets collected from OpenAI Acrobot, CartPOle and MountainCar environments Brockman et al. (2016) using DQN features. To collect these features, we trained 1 hidden layer neural network with DQN algorithm Mnih et al. (2015) for 1000 plays. Then, the trained agent played 5000 episodes following greedy policy, while neural network hidden states were recorded as feature representation of the visited states. Features collected this way tend to be highly correlated, therefore we applied PCA clearing, keeping minimum set of principal components, corresponding to 90 % of the variance.

For the TD-SVRG algorithm we used theoretically justified parameters, for the other algorithms parameters selected with grid search (Sec. J.2), the results are presented in Figure 5. In all environments TD-SVRG exhibits stable linear convergence, GTD2 and vanilla TD algorithms converge sublinearly, while PD-SVRG performance is unstable due to high range of condition numbers of dataset's characteristic matrices $A$ and $C$ (large values of $\kappa(C)$ caused PD-SVRG divergence in the Acrobot dataset).

### J.4 Batched SVRG performance

In this set of experiments we compare the performance of TD-SVRG and batched TD-SVRG in the finite-sample case. We generate 10 datasets of size 50000 from a similar MDP as in Section 6. Algorithms run with

the same hyperparameters. Average results over 10 runs are presented in Figure 6 and show that batched TD-SVRG saves a lot of computations during the earlier epochs, which provides faster convergence.

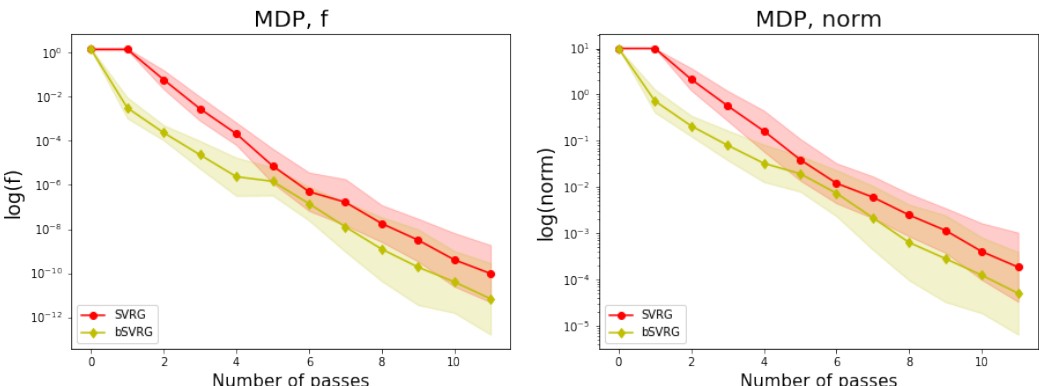

Figure 6: Average performance of TD-SVRG and batching TD-SVRG in the finite sample case. Datasets sampled from MDP environments. Left figure – performance in terms of $\log(f(\theta))$. Right figure – performance in terms of $\log(|\theta - \theta^*|)$.

## J.5 Online i.i.d. sampling from the MDP

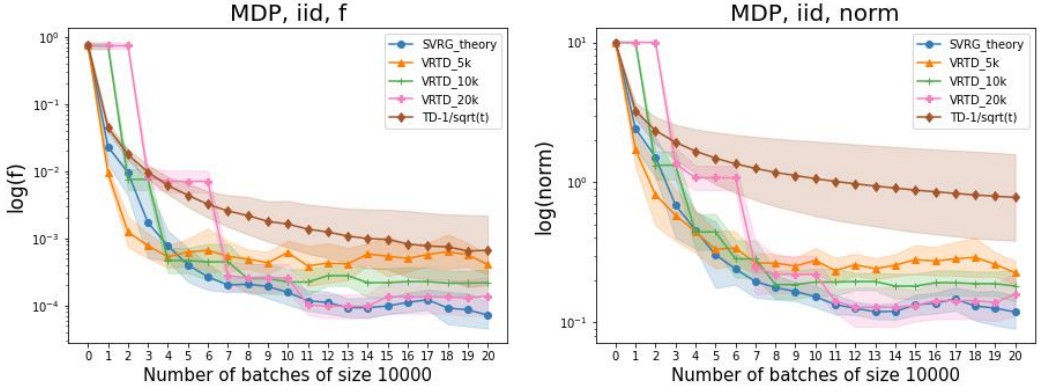

Figure 7: **Online iid sampling:** Average performance of TD-SVRG with theoretical parameters, VRTD with different batch sizes and vanilla TD with learning rate equal to $1/\sqrt{t}$ in the i.i.d. sampling case. Left figure – performance in terms of $\log(f(\theta))$, right figure in terms of $\log(|\theta - \theta^*|)$.

In this set of experiments we compare the performance of TD-SVRG, VRTD and Vanilla TD with decreasing learning rates in the i.i.d. sampling case. States and rewards are sampled from the same MDP as in Section 6 under iid sampling strategy - next transition is being sampled independently from previous transition. Hyperparameters are chosen as follows: for TD-SVRG – learning rate $\alpha = 1/8$, update batch size $M = 16/\lambda_A$, estimation batch size epoch expansion factor is $\rho^2 = 1.2$. VRTD – learning rate $\alpha = 0.1$ and batch sizes $M \in 5, 10, 20 * 10^3$. For vanilla TD decreasing learning rate is set to $1/\sqrt{t}$, where $t$ is a number of the performed update. Average results over 10 runs are shown in Figure 7. The figure shows that TD-SVRG converges even if its performance suffers from high variance, VRTD algorithms oscillate after reaching a certain level (due to bias). Vanilla TD with decreasing learning rate converges slowly then SVRG.

### J.6   Online Markovian sampling from an MDP

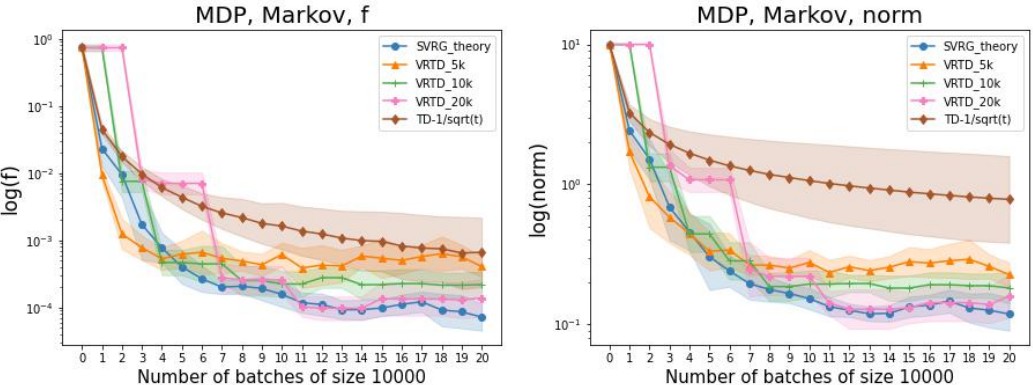

Figure 8: **Online Markovian sampling:** Average performance of TD-SVRG with theoretical parameters, VRTD with different batch sizes and vanilla TD with learning rate equal to $1/\sqrt{t}$ in the i.i.d. sampling case. Left figure – performance in terms of $\log(f(\theta))$, right figure in terms of $\log(|\theta - \theta^*|)$.

In this set of experiments we compare the performance of TD-SVRG, VRTD and Vanilla TD with decreasing learning rates in the Markovian sampling case. States and rewards are sampled from the same MDP as in Section 6 under Markovian sampling strategy - next transition is being sampled dependent on the previous transition. Hyperparameters are chosen as follows: for TD-SVRG – learning rate $\alpha = 1/8$, update batch size $M = 16/\lambda_A$, estimation batch size epoch expansion factor is $\rho^2 = 1.2$. VRTD – learning rate $\alpha = 0.1$ and batch sizes $M \in 5, 10, 20 * 10^3$. For vanilla TD decreasing learning rate is set to $1/\sqrt{t}$, where $t$ is a number of the performed update. Average results over 10 runs are shown in Figure 8. Because this MDP mixes very fast even under Markovian sampling, the results are very similar to iid sampling case. The figure shows that TD-SVRG converges with decreasing rate, VRTD algorithms reach certain level and then oscillate, vanilla TD converges with decreasing rate and slower then TD-SVRG.

### J.7   Comparison of update batch sizes

In this set of experiments, we assume that $\lambda_A$ and, consequently, the theory-predicted batch size $16/\lambda_A$ are not known. We investigate the effect of approximate update batch size on the algorithm's performance, checking the performance of batch sizes $\{8, 12, 24, 32\}/\lambda_A$ against the theory-predicted value. We run this experiment for all three cases: dataset (Figure 9), online iid sampling (Figure 10), and online Markovian sampling (Figure 11). In all three cases, the algorithms demonstrate comparable performance, while in the online sampling cases, both iid and Markovian, the difference is negligible. This is caused by the fact that mean-path update estimation dominates the complexity.

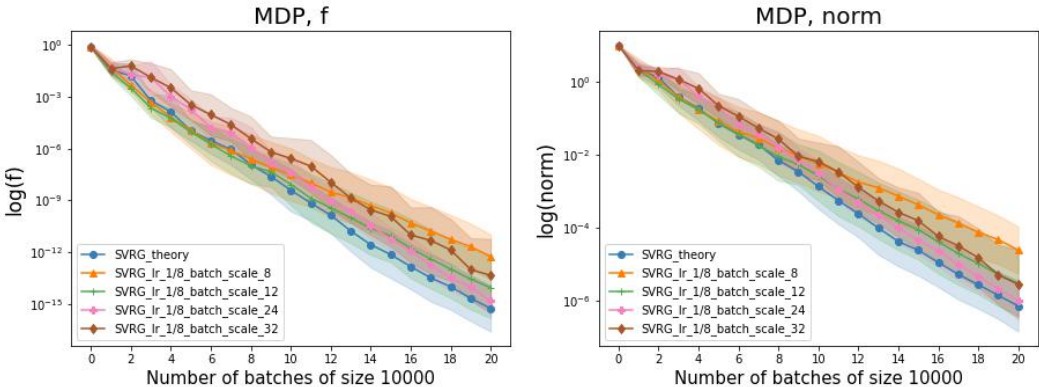

Figure 9: **Dataset sampling case:** Average performance of TD-SVRG with theoretical parameters and with different update batch size scales of $1/\lambda_A$.

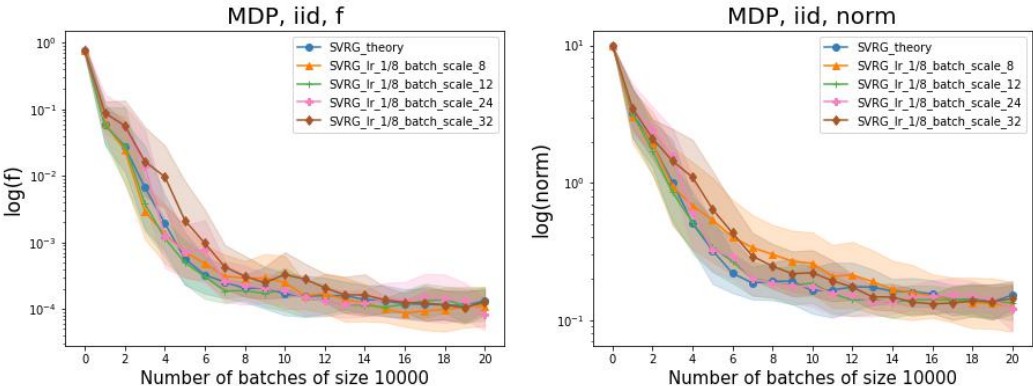

Figure 10: **IID sampling case:** Average performance of TD-SVRG with theoretical parameters and with different update batch size scales of $1/\lambda_A$.

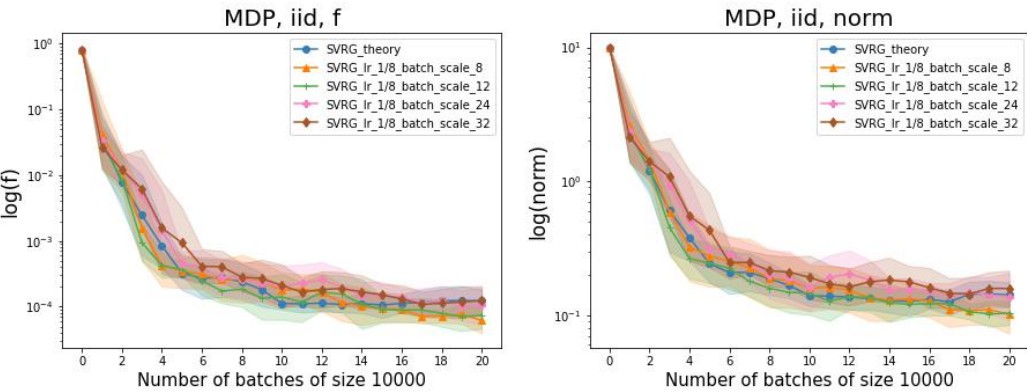

Figure 11: **Markovian sampling case:** Average performance of TD-SVRG with theoretical parameters and with different update batch size scales of $1/\lambda_A$.

### J.8 Experiment details

In our experiments (Section 6) we compare the performance of TD-SVRG with GTD2 Sutton et al. (2009), "vanilla" TD learning Sutton (1988), and PD-SVRG Du et al. (2017) in the finite sample setting. Generally, our experimental set-up is similar to Peng et al. (2020). Datasets of size 5,000 are generated from 4 environments: Random MDP Dann et al. (2014), and the Acrobot, CartPole and Mountain car OpenAI Gym environments Brockman et al. (2016). For the Random MDP, we construct an MDP environment with $|S| = 400$, 21 features (20 random and 1 constant) and 10 actions, with action selection probabilities generated from $U[0, 1]$. For OpenAI gym environments, the agent selects states uniformly at random. Features are constructed by applying RBF kernels to discretize the original states and then removing highly correlated features. The decay rate $\gamma$ is set to 0.95.

We compare the performance of TD-SVRG against the performance of other algorithms with parameters selected by grid search. Details on the grid search might be found in Appendix J.2. Hyperparamters for the algorithms are selected as follows: for TD-SVRG our theoretically justified parameters are selected, the learning rate is set to $\alpha = 1/8$ and the update batch size to $M = 16/\lambda_A$; for GTD2 the best performing parameters were: $\alpha = 0.125$ and $\beta = 0.25$; for vanilla TD a decreasing learning rate is set to $\alpha = 1/\sqrt{t}$; for PD-SVRG the parameters are set to $\sigma_\theta = 0.1/(L_\rho \kappa(\hat{C}))$, $\sigma_w = 0.1/\lambda_{\max}(C)$ and the batch size is twice the size of the dataset, i.e., $M = 2N$. Each algorithm for each setting was run 10 times and the geometric average performance is presented.

## K   Algorithms comparison

In this section, we present a more detailed comparison of TD algorithms. Our results are summarized in Table 6, and a detailed explanation of the quantities in the table is provided below.

Please note that while other algorithms derive convergence in terms of $||\theta - \theta^|||^2$, our convergence is expressed in terms of the function $f(\theta)$. The results can be compared using the inequality $\lambda_A ||\theta - \theta^|||^2 \leq f(\theta) \leq ||\theta - \theta^|||^2$. This implies that achieving an accuracy of $\epsilon$ in terms of one quantity can be accomplished by achieving an accuracy of $\lambda_A/\epsilon$ in terms of the other quantity. Consequently, our results for the finite sample case are strictly superior. For environment sampling cases, our results imply previous findings, whereas our results are not implied by previous ones. Furthermore, it is worth noting that the inequality $\lambda_A ||\theta - \theta^|||^2 \leq f(\theta)$ is rarely strict, which means that in most cases, the convergence implied by our results would be superior.

Table 6: Comparison of algorithmic parameters. PD-SVRG and PD SAGA results reported from Du et al. (2017), VRTD and TD results from Xu et al. (2020), GTD2 from Touati et al. (2018). $\lambda_{\min}(Q)$ and $\kappa(Q)$ are used to define, respectively, minimum eigenvalue and condition number of a matrix $Q$. $\lambda_A$ in this table denotes minimum eigenvalue of the matrix $1/2(A + A^T)$, which is defined in Equation (1). Finite sample results use $N$ for the size of the dataset sampled from the MDP. Other notation is taken from original papers, and Section 1 in the supplementary information gives self-contained definitions of all the symbols appearing in this table. For simplicity $1 + \gamma$ is upper bounded by 2 throughout, where $\gamma$ is the discount factor.

| Method | Learning rate | Batch size | Total complexity |
|---|---|---|---|
| Finite sample case | | | |
| GTD2 | $\frac{9^2 \times 2\sigma}{8\sigma^2(k+2) + 9^2\zeta}$ | 1 | $\mathcal{O}\left(\frac{\kappa(Q)^2 \mathcal{H}}{\lambda_{\min}(G)\epsilon}\right)$ |
| PD-SVRG | $\frac{\lambda_{\min}(A^T C^{-1} A)}{48\kappa(C)L_G^2}$ | $\frac{51\kappa^2(C)L_G^2}{\lambda_{\min}(A^T C^{-1} A)^2}$ | $\mathcal{O}\left(\left(N + \left(\frac{\kappa^2(C)L_G^2}{\lambda_{\min}(A^T C^{-1} A)^2}\right)\right)\log(\frac{1}{\epsilon})\right)$ |
| PD SAGA | $\frac{\lambda_{\min}(A^T C^{-1} A)}{3(8\kappa^2(C)L_G^2 + n\mu_\rho)}$ | 1 | $\mathcal{O}\left(\left(N + \frac{\kappa^2(C)L_G^2}{\lambda_{\min}(A^T C^{-1} A)^2}\right)\log(\frac{1}{\epsilon})\right)$ |
| This paper | $1/8$ | $16/\lambda_A$ | $\mathcal{O}\left(\left(N + \frac{1}{\lambda_A}\right)\log(\frac{1}{\epsilon})\right)$ |
| i.i.d. sampling | | | |
| TD | $\min(\frac{\lambda_A}{4}, \frac{1}{2\lambda_A})$ | 1 | $\mathcal{O}\left(\frac{1}{\epsilon\lambda_A^2}\log(\frac{1}{\epsilon})\right)$ |
| This paper | $1/8$ | $16/\lambda_A$ | $\mathcal{O}\left(\frac{1}{\epsilon\lambda_A}\log(\frac{1}{\epsilon})\right)$ |
| Markovian sampling | | | |
| TD | $\mathcal{O}(\epsilon/\log(\frac{1}{\epsilon}))$ | 1 | $\mathcal{O}\left(\frac{1}{\epsilon\lambda_A^2}\log^2(\frac{1}{\epsilon})\right)$ |
| VRDT | $\mathcal{O}(\lambda_A)$ | $\mathcal{O}\left(\frac{1}{\epsilon\lambda_A^2}\right)$ | $\mathcal{O}\left(\frac{1}{\epsilon\lambda_A^2}\log(\frac{1}{\epsilon})\right)$ |
| This paper | $\mathcal{O}(\epsilon/\log(\frac{1}{\epsilon}))$ | $\mathcal{O}\left(\frac{\log(\frac{1}{\epsilon})}{\epsilon\lambda_A}\right)$ | $\mathcal{O}\left(\frac{1}{\epsilon\lambda_A}\log^2(\frac{1}{\epsilon})\right)$ |

Definitions of quantities in Table 6:

**GTD2** convergence analysis resutls are taken from Touati et al. (2018). The learning rate required for their guarantee to work is set to $\frac{9^2 \times 2\sigma}{8\sigma^2(k+2) + 9^2\zeta}$ and the complexity to obtain accuracy $\epsilon$ is $\mathcal{O}(\frac{\kappa(Q)^2 \mathcal{H}d}{\lambda_{\min}(G)\epsilon})$. In this notation:

- $\sigma$ is the minimum eigenvalue of the matrix $A'^T M^{-1} A'$, where the matrix $M = \mathbb{E}[\phi(s_k, a_k)\phi(s_k, a_k)^T]$ and $A' = \mathbb{E}[e_k(\gamma \mathbb{E}_\pi[\phi(s_{k+1}, .] - \phi(s_k, a_k))^T]$, where $e_k$ is the eligibility trace vector $e_k = \lambda\gamma\kappa(s_k, a_k)e_{k-1} + \phi(s_k, a_k)$.

- $k$ is an iteration number.

- The matrix $G$ plays key role in the analysis, it is a block matrix of the form

$$G = \begin{pmatrix} 0 & \sqrt{\beta}A'^T \\ -\sqrt{\beta}A' & \beta M_k \end{pmatrix},$$

  and $G_k$ is a matrix of similar form generated from quantities estimated at time point $k$.

- $\zeta$ is $2 \times 9^2 c(M)^2 \rho^2 + 32 c(M) L_G$, where $c(M)$ is the condition number of the matrix $M$, $\rho$ is the maximum eigenvalue of the matrix $A'^T M^{-1} A'$ and $L_G$ is the $L_G = ||\mathbb{E}[G_K^T G_K | \mathcal{F}_{k-1}]||$. $\mathcal{F}_{k-1}$ in this analysis is the $\sigma$-algebra generated by all previous history up to moment $k-1$.

- The quantity $\mathcal{H}$ is equal to $\mathbb{E}||G_K z^* - g_k||$, where $z^* = (\theta^*, \frac{1}{\sqrt{\beta}w^*})$ is the optimal solution and $g_k = (0, \frac{1}{\sqrt{\beta}}b)$.

- The last quantity left undefined is $\kappa(Q)$, which is the condition number of the matrix $Q$, obtained by diagonalization of the matrix $G = Q^T \Lambda Q$.

**PD-SVRG** and **PD SAGA** use the same quantities as **GTD2**, except that matrices $A$ and $C$ are defined the same way as in this paper: $A = \mathbb{E}[(\phi(s)^T - \gamma\phi(s')^T)\phi(s)]$, $C = \mathbb{E}[\phi(s)\phi^T(s)]$.

- $n$ in this notation is the size of the dataset.

- $\mu_\rho$ is the minimum eigenvalue of matrix $A^T C^{-1} A$.

All other quantities are defined in the paper.

