# OpenReview forum: "Closing the gap between SVRG and TD-SVRG with Gradient Splitting"
_TMLR — Accepted by TMLR_

### Review · Reviewer_mVeV · 2024-04-19

**Summary Of Contributions:**

This paper focuses on policy evaluation in reinforcement learning. The paper shows improved convergence rates of TD-learning combined with variance reduction techniques (SVRG), in three different settings (finite, online i.i.d, and online Markovian), when compared with the existing results in the literature.

**Audience:**

Yes

**Broader Impact Concerns:**

I do not predict relevant ethical implications of the work in particular.

**Claims And Evidence:**

No

**Requested Changes:**

I believe it is important that the following points are addressed, with critical impact on my recommendation:
- Clarification of the novelty of the algorithm;
- Adding the derivation of the algorithm, or an explanation on how the variance is reduced;
- Interpretation of the algorithm, and definition of the algorithm in the non-finite-sample case;
- Definition of the notion of complexity used in Table 1;
- Clarification on the experimental results, specifically what is considered an update;
- Experimental results in terms of computation time.

**Strengths And Weaknesses:**

The contribution of the work is strong by lowering the complexity of TD-learning when compared with the existing algorithms. The theoretical results are confirmed by clear improvement over the baselines in the experimental section. Finally, the variance reduction and gradient splitting techniques used to establish the algorithm and theory appear to be non-trivial and may inspire further work on the broader area.

However, I believe the paper is not clear in a few important aspects, impacting the evaluation of the work. Specifically, at the moment, I am unclear about the following:
- Is the algorithm new, or only the theoretical analysis, resulting in improved rates for the same algorithm?
- The algorithm is not explained, nor introduced, and I have some questions. Does the algorithm come in a straightforward way from a variance reduction technique? Why is the technique not described, nor how the improvements come into place?
- In the finite-sample case, the algorithm (Algorithm 1) has epochs, where an entire dataset is used to average the TD errors, and inside each epoch, has M iterations, where M samples are used (one per iteration). The complexity mentioned in the paper (for instance, table 1), is with respect to the number of epochs, iterations, or samples?
- How does the algorithm work in the non-finite-sample case? Could a pseudo-code be also added, or Algorithm 1 written in a general fashion?
- In the experimental results, we see clear benefits of the approach in terms of number of updates. Is an iteration of TD-SVRG an epoch or an iteration? And since each of these uses more samples (the entire dataset), can we see the results in terms of computation time?
- What is meant by geometric convergence? On the number of epochs/iterations?

I also have the following less important comments and questions:
- Typo: "worsk" (page 9);
- Was "linearly" meant "geometrically" on Section 6, of the experiments?
- The "condition number" is not defined;
- I believe $\bar{g}_m$ should be $\bar{g}_m(\bar(\theta))$ on the inner cycle of Algorithm 1;
- If throughout the inner cycle of Algorithm 1, $\bar{g}_m$ is constant, and given the linearity of the updates, why add it on every update and not just after the cycle, possibly with a different learning rate?

---

> ### Author Response · Authors · 2024-05-10
> **Answer to Rewiever mVeV**
>
> We would like to thank the reviewer for their careful review and positive evaluation of our contributions. We next answer your questions and concerns:
>
> - **Is the algorithm new, or only the theoretical analysis, resulting in improved rates for the same algorithm? Does the algorithm come in a straightforward way from a variance reduction technique?**
>
>  The real novelty of this paper is in the theoretical analysis, however the algorithm is also new:  we are not aware of any papers that have previously considered it. However, it  is also the most straightforward way to apply the variance reduction technique. To the extent it was not introduced in the previous literature, it is likely because this method is more challenging to analyze.
>
> - **The algorithm is not explained, nor introduced, and I have some questions.  Why is the technique not described, nor how the improvements come into place?**
>
> Please see Section 3 in the revised manuscript where we introduce the core idea behind the variance reduction technique (blue text).
>
> - **In the finite-sample case, the algorithm (Algorithm 1) has epochs, where an entire dataset is used to average the TD errors, and inside each epoch, has M iterations, where M samples are used (one per iteration). The complexity mentioned in the paper (for instance, table 1), is with respect to the number of epochs, iterations, or samples?**
>
> What is reported there is the  number of updates. This is a slightly different -- and better --  measure than number of iterations since an ``update'' includes both iteration in the inner loop as well as the updates to compute the mean value $\bar{g}_m(\theta)$ before the inner loop is started.  In the environment cases (i.i.d + Markovian), this  is identical to sample complexity (up to constants) since one is generating new samples in each step of the inner loop. In the dataset cases, this is not sample complexity since one then makes many passes through the same dataset.
>
> - **How does the algorithm work in the non-finite-sample case? Could a pseudo-code be also added, or Algorithm 1 written in a general fashion?**
>
> This is present in the appendix of the paper. Indeed, the algorithms in both finite and non-finite sample cases exhibit considerable similarity, leading us to the decision not to include other versions of the algorithms in the main body of the paper. Instead, we provide a concise description of the differences within the main text, while the pseudocode for the algorithm in these alternate cases can be found in the Appendix: algorithm for batched TD-SVRG in Appendix F, for iid sampling case in Appendix G and for Markovian sampling case in Appendix H.
>
> - **In the experimental results, we see clear benefits of the approach in terms of number of updates. Is an iteration of TD-SVRG an epoch or an iteration? And since each of these uses more samples (the entire dataset), can we see the results in terms of computation time?**
>
> We present our results in terms of the number of update computations, as mentioned in our response to the previous question. This metric corresponds to computational time.
>
> - **Was "linearly" meant "geometrically" on Section 6, of the experiments?**
>
> Yes. We used the terms "linear convergence" and "geometric convergence" interchangeably. To prevent confusion, we will substitute "linearly" with "geometrically" throughout the entire paper.
>
> - **The "condition number" is not defined**
>
> By 'condition number,' we refer to the ratio of the maximum to minimum eigenvalues of a matrix. We have put a definition into  the manuscript, see the blue text in Subsection "Motivation and Contribution", where this term appears for the first time.
>
> - **I believe $\bar{g}_m$ should be  $\bar{g}_m (\tilde{\theta})$ on the inner cycle of Algorithm 1**
>
> We have changed $\bar{g}_m$ to $\bar{g}_m (\tilde{\theta})$ in the inner loop of the algorithm to avoid the confusion.
>
> -**If throughout the inner cycle of Algorithm 1, $\bar{g}_m$ is constant, and given the linearity of the updates, why add it on every update and not just after the cycle, possibly with a different learning rate?**
>
> This doesn't work because throughout the inner cycle, we use  $g_{s,s'}(\cdot)$ evaluated at points obtained earlier in the cycle. In slightly more detail: at each iteration $t$, the update $g_{s,s'}(\theta_{t-1})$ depends on the parameters obtained during the previous iteration. If $\bar{g}_m$
>
> was not added in the previous iteration, the parameter vector
> $\theta_{t-1}$ would be different.

---

> ### Comment · Reviewer_mVeV · 2024-07-12
> **New comment**
>
> I cannot see a response from the authors to my review. Even though I can see that some things were changed in the pdf submission (some text is highlighted in blue), and that some of those things relate to my comments (some of them also intersect with other reviewers' comments) requiring clarifications, it is hard to evaluate exactly what is addressed and what is not, in the paper, and why, without a tailored response. Therefore, even though some of my concerns may have been addressed, it will be difficult to have them into account in the official recommendation.

---

> > ### Author Response · Authors · 2024-07-12
> > **Reply**
> >
> > Dear Reviewer,
> >
> > Apologies, it seemed that we did not set permissions on our reply appropriately. It should be visible now (beginning with the words "We would like to thank the reviewer for their careful review..."). Please let us know if any issues persist.

---

> > > ### Comment · Reviewer_mVeV · 2024-07-12
> > > **Update**
> > >
> > > I can see the response, now. Thank you for the response.
> > >
> > > I see the authors clarified all of my questions satisfactorily, and have also updated the manuscript accordingly.

---

### Review · Reviewer_Pfqp · 2024-05-27

**Summary Of Contributions:**

The primary contribution of this paper lies in leveraging the gradient splitting interpretation of the Temporal Difference (TD) algorithm to apply Stochastic Variance Reduced Gradient (SVRG), achieving a linear convergence rate. By adopting the gradient splitting perspective, the authors formulate a quadratic objective function whose directional derivative aligns with the TD gradient when the direction points toward the optimal parameter. This insight allows for an analysis of the convergence of SVRG in optimizing the quadratic function, yielding improved dependence on problem-specific parameters. The authors examine three distinct sampling regimes: i.i.d. setting, Markovian setting, and finite data setting. In the finite data setting, akin to finite sum minimization, they establish a geometric convergence rate. For the i.i.d. and Markovian settings, they demonstrate a sublinear convergence rate due to the computational burden of evaluating the full gradient.

**Audience:**

Yes

**Broader Impact Concerns:**

There is no concern.

**Claims And Evidence:**

Yes

**Requested Changes:**

1- To set the step-size or batch size according to your theory, one needs to know the minimum eigenvalue of matrix A. But in some settings knowing that parameter is computationally expensive. How do you estimate that? It would be nice to see the effect of misestimation of this value empirically.

2-  There are some parameter overloading like m used as an iteration index or a constant. There are some parameters defined which is not used in the algorithm such as s in instantiation of Alg 4. So the paper needs a polish in terms of parameter definition and reuse.

**Strengths And Weaknesses:**

Getting better dependence on the problem-dependent constants compared to the previous works is the main strength of this work.

The paper needs restructuring. Mainly the paper leverages gradient splitting to get the desired objective function and convergence rates but it explains that in section 5.  The experimental part in the main body just considers the finite dataset case and ignores the other two online sampling versions.

---

> ### Author Response · Authors · 2024-06-26
> **Answer to reviewer Pfqp**
>
> We would like to thank the reviewer for their careful review and positive evaluation of our contributions. We next answer your questions and concerns:
>
> - **The paper needs restructuring. Mainly the paper leverages gradient splitting to get the desired objective function and convergence rates but it explains that in section 5. The experimental part in the main body just considers the finite dataset case and ignores the other two online sampling versions.**
>
> The requested changes have been incorporated into the final version of the manuscript. In particular, we Sections 4 and 5 were swapped. We did not include the online sampling experiments due to space constraints. Since it is a theoretical paper, we prioritized theory in the main body of the paper and adding additional experiment result figure takes a lot of space.
>
> - **To set the step-size or batch size according to your theory, one needs to know the minimum eigenvalue of matrix A. But in some settings knowing that parameter is computationally expensive. How do you estimate that? It would be nice to see the effect of misestimation of this value empirically.**
>
> This is an open question in the literature:  all previous works (including our own) require the knowledge of this eigenvalue to achieve geometric convergence. Our result is an improvement over other results under the same setting but does not resolve this open problem.
>
> As to the effect of misestimation, we next sketch out an upper bound that shows that if the correct estimation batch size is $\beta N$ while we guess $\kappa N$, then the number of updates is amplified by $\max(\beta/\kappa,1)$.
> Indeed,  note that the average number of samples and updates required to perform one epoch in our algorithm to achieve the expected contraction of $\rho$, is equal to $\beta N/2 + N$ (where again $\beta N$ is the estimation batch size).
> If $\kappa$ is bigger than $\beta$, we are doing more updates than necessary and one can show we will achieve the expected $\rho$-contraction in one iteration but at the cost of $\kappa N + N$. But if $\kappa < \beta$, we can use the fact that the number of updates we perform in the inner loop is random between $0$ and $\beta N$; the chance that the randomly sampled number of updates  falls under $\kappa$ and guarantees an expected update is $\kappa / \beta$. Therefore, we need on average $\beta / \kappa$ epochs to get the same $\rho$-contraction.
>
> - **There are some parameter overloading like m used as an iteration index or a constant. There are some parameters defined which is not used in the algorithm such as s in instantiation of Alg 4. So the paper needs a polish in terms of parameter definition and reuse.**
>
> We apologize for the confusion with the notation, both of them are related to the same problem. We will unify notation through the paper: $m'$ is an epoch index and $m$ is a total number of epochs.

---

### Review · Reviewer_HR3f · 2024-06-13

**Summary Of Contributions:**

This paper uses the finding of Liu and Olshevsky (2021) that temporal difference updates (in the linear, policy evaluation setting) can be viewed as a gradient splitting of a quadratic objective function to analyze the convergence rate of the Stochastic Variance Reduced Gradient method applied to TD learning. The result is an analysis showing geometric convergence rate for a simple algorithm which applies SVRG to the TD update direction. The algorithm is further shown to exhibit faster convergence than standard TD methods along with other variance-reduction approaches on a variety of environments from the OpenAI Gym suite.

**Audience:**

Yes

**Claims And Evidence:**

No

**Requested Changes:**

- Please address the concerns raised in 'weaknesses'.
- Please review the appropriate use of \citep and \citet

**Strengths And Weaknesses:**

**Strengths**

- The key ideas put forward in the paper is simple and clearly presented
- The assumptions required for the results to hold are clearly stated, and effort is made to provide weaker results when certain assumptions (e.g. the balanced dataset assumption) are violated
- The algorithm is clearly presented and easy to understand.
- The use of gradient splitting, while simple, makes the analysis of the variance-reduced TD algorithm clean and obtains much better convergence results than the cited literature.

**Weaknesses**
1. While its simplicity makes the paper easy to follow, this simplicity is a symptom of the relative lack of novelty in this paper, which essentially investigates a corollary of the finding of Liu and Olshevsky (who note that the gradient splitting approach can be applied to analyze a number of variants of TD using similar tools as for gradient descent). Although my field of expertise isn't optimization and so I may have missed something, it doesn't appear that the paper provides new technical tools for studying the convergence of variance reduction methods in TD learning.
2. The exponential convergence result requires sampling the full dataset and computing a full-batch gradient for every increment of $m$ (or in the batched case, sampling an exponentially growing subset of the dataset). This seems like it should add significantly to the computational cost of the algorithm, and it would be helpful to understand how the loss-reduction-per-flop compares with other methods, rather than loss-reduction-per-iteration (where each iteration has an exponentially growing computational cost). This is particularly relevant for the empirical evaluations, where it isn't clear whether the x-axis is providing a fair comparison between algorithms. How does TD-SVRG perform in terms of wall-clock time?
3. Some of the quantities in theorems/equations are not always clearly defined. For example, what does it mean that $c$ is a parameter? Is its optimal value instance-dependent or instance-independent? Expectations are also often not qualified, so it is not clear what distribution and what random variable the expectation is being taken over.
4. In the proof of Lemma 1, the quantity $\bar{g}(\theta^*)$ appears out of nowhere and is not justified in the text that follows its appearance.
5. The objective for which the td update g is a gradient splitting doesn't appear to relate in an obvious way to the bellman error / projected bellman error / value error. It would be helpful to understand how the convergence of the parameters / auxiliary objective translates to the convergence of the TD objective.

---

> ### Author Response · Authors · 2024-06-26
> **Answer to reviewer HR3f**
>
> We would like to thank the reviewer for positive evaluation of our contribution and noting the clarity of our presentation, we made an extra effort to make the paper clear and easy to follow. We put an extra effort to make this theoretical paper clear, easy to read and, thus, interesting to wider audience. We next answer your questions and concerns.
>
> - **While its simplicity makes the paper easy to follow, this simplicity is a symptom of the relative lack of novelty in this paper, which essentially investigates a corollary of the finding of Liu and Olshevsky (who note that the gradient splitting approach can be applied to analyze a number of variants of TD using similar tools as for gradient descent). Although my field of expertise isn't optimization and so I may have missed something, it doesn't appear that the paper provides new technical tools for studying the convergence of variance reduction methods in TD learning.**
>
> Our response here has three parts.  **First,**  it is not quite correct  that the paper investigates a corollary of the previous work by Liu and Olshevsky (2021). While the latter paper is important conceptually, we do not use the techniques from that paper in our proof, nor do we use the main finding, which is the exact form of the objective being "split" by TD:
> $$ f(\theta) = (1-\gamma)||V_\theta-V^*||_D^2 + \gamma||V_\theta-V^*||_{Dir}^2.$$
> We mention this but never actually use it (e.g., we leave it as $f(\theta)$ throughout our analyses). In particular, our proof would need only slight modifications to work with TD analyses which predate Liu and Olshevsky (2021), which used the lower bounds on the inner product $v_t(\theta)^T (\theta - \theta^*)$ (where $\theta^*$ is the TD fixed point and $v_t$ is the SVRG update). Rather, the gradient splitting interpretation is an important motivation for us which gives strong indications for what kind of relations should/should not be true -- it is why we were able to guess the right way to "port" the classic SVRG analysis of Johnson \& Yang to the TD setting (Lemma 5.1 and its variations).
>
> **Second**, the history of the previous results on this issue serves as a good demonstration of the problem complexity: the first analysis was done by Korda and La (2015), but the Markovian sampling part of it had an error. Their results were reanalyzed in Xu et al (2020) and improved in Ma et al (2020), but they, in turn, made an error in their analysis of the iid sampling case. Tackling these issues required a lot of work and new findings, which were not presented in the Liu and Olshevsky paper and were derived from scratch. The final result may appear simple, but it is clearly not the kind of simplicity that was apparent to researchers in the area from the start.
>
> **Third**, we would refer the reviewer to the TMLR acceptance criteria (\url{https://www.jmlr.org/tmlr/acceptance-criteria.html}) specifically:
>
> *Crucially, it should not be used as a reason to reject work that isn't considered “significant” or “impactful” because it isn't achieving a new state-of-the-art on some benchmark. Nor should it form the basis for rejecting work on a method considered not “novel enough”, as novelty of the studied method is not a necessary criteria for acceptance. We explicitly avoid these terms (“significant”, “impactful”, “novel”), and focus instead on the notion of “interest”. If the authors make it clear that there is something to be learned by some researchers in their area from their work, then the criterion of interest is considered satisfied. TMLR instead relies on certifications (such as “Featured” and “Outstanding”) to provide annotations on submissions that pertain to (more speculative) assertions on significance or potential for impact.*
>
> Given that, in their review, the reviewer has acknowledged that our results are "simple," "clearly presented," "clean," and obtain "much better convergence results than the cited literature," we submit that this criterion is clearly satisfied.

---

> ### Author Response · Authors · 2024-06-26
> **Answer to reviewer HR3f part 2**
>
> - **The exponential convergence result requires sampling the full dataset and computing a full-batch gradient for every increment of $m$ (or in the batched case, sampling an exponentially growing subset of the dataset). This seems like it should add significantly to the computational cost of the algorithm, and it would be helpful to understand how the loss-reduction-per-flop compares with other methods, rather than loss-reduction-per-iteration (where each iteration has an exponentially growing computational cost). This is particularly relevant for the empirical evaluations, where it isn't clear whether the x-axis is providing a fair comparison between algorithms. How does TD-SVRG perform in terms of wall-clock time?**
>
> We indeed give a fair comparison because we count basic updates, e.g., $g_{s,s'}(\theta)$, a TD update for sampled pair of states $s,s'$ and parameter vector $\theta$ is a basic update.
>
> Likewise, computing
> \bar$g_m= \frac{1}{N} \sum_{s,s'} g_{s,s'} (\tilde{\theta})$
> counts as $N$ basic updates since one has to compute $N$ quantities $g_{s,s'}$.
> In our experiments, reported number of basic updates includes both basic updates required to execute one iteration and basic updates needed to estimate mean-path update. Therefore, the $x$-axis in our Figures is very close to wall-time (which each basic update might take a little longer or a little less during execution, this will average out over the course of the algorithm).
> We have added this clarification to the Experimental results section of the paper.
>
> - **Some of the quantities in theorems/equations are not always clearly defined. For example, what does it mean that $c$ is a parameter? Is its optimal value instance-dependent or instance-independent? Expectations are also often not qualified, so it is not clear what distribution and what random variable the expectation is being taken over.**
>
> We can choose $c$ to be anything: it increases the accuracy of mean path estimation at the expense of total number of iterations. It is independent on $\epsilon$ or $\lambda_A$, two quantities in which we measure complexity. Its optimal value is instant dependent, but cannot be chosen during the run of the algorithm, since the quantities required to make this choice are not known.
>
> We read the manuscript and found that every expectation is defined in the text, but to improve readability we edited the notation. The expectation with respect to all randomness during the run of the algorithm is denoted as $\mathbb{E}$. Expectation with respect to a randomly sampled pair of states ${s,s'}$ denoted as $\mathbb{E}_{s,s'}$. Expectation with respect to any other random events denoted as an expectation conditioned on sigma field $\mathbb{E}[\cdot |\mathcal{F} ]$, with sigma field being specified in the text.
>
> - **In the proof of Lemma 1, the quantity  \bar{g}$(\theta)$ appears out of nowhere and is not justified in the text that follows its appearance.**
>
> Let us politely disagree here: the quantity \bar{g}$(\theta)$ - mean path updated as a function of a parameter vector $\theta$ - is one of the key quantities of our paper. It is defined in Equation 1 in the problem formulation section. To avoid the confusion, we have added a reference to the definition to the proof of Lemma 1.

---

> > ### Author Response · Authors · 2024-06-26
> > **Answer to reviewer HR3f part 3**
> >
> > - **The objective for which the td update g is a gradient splitting doesn't appear to relate in an obvious way to the bellman error / projected bellman error / value error. It would be helpful to understand how the convergence of the parameters / auxiliary objective translates to the convergence of the TD objective.**
> >
> > There are some related bounds with this flavor in the supplementary information of Liu and Olshevsky (2021) relating the gradient splitting objective to the more usual weighted L2 norm used for TD analysis.
> >
> > Indeed, the standard analysis of TD learning uses the weighted two-norm
> > $$ D(x) = \sum_{s} \mu(s) x(s)^2,$$ where $\mu$ is the stationary distribution corresponding to the policy being evaluated.  Usually, bounds for convergence of time of TD-learning are in terms of this norm.
> >
> > We can try to relate the gradient splitting objective to this familiar quantity. Indeed, the gradient splitting objective contains this $D$-norm, but also has a Dirichlet seminorm
> > $$ Dir(x) = \sum_{s,s'} \mu(s) P(s'|s) (x(s)-x(s'))^2.$$
> > This is a seminorm since it equals zero when $x(s)$ is a multiple of the one vector, and can be thought of as measuring the projection of a vector onto ${\bf 1}^\perp$. For such a vector $x \in {\bf 1}^\perp$, one can bounds
> > $$ Dir(x) \leq r D(x),$$ where $r$ is written out explicitly in Lemma 2 of Liu and Olshevsky (2021). To avoid confusion, please note once again that this only holds for $x \in {\bf 1}^\perp$.
> >
> > Further, it is then possible to decompose
> > $$ D(x-y) = (m_x - m_y)^2 + D(x_{\bf 1^\perp} - y_{\bf 1^\perp}) \leq (m_x - m_y)^2 \leq (m_x - m_y)^2 + r Dir(x-y), $$ where $m_x$ is the weighted mean of the vector $x$:
> > $$ m_x = \sum_{s} \mu(s) x(s). $$
> >  This is quite similar to Eq. (28) of Liu and Olshevsky (2021). The gradient splitting objective can be used to bound the last $rDir(x-y)$ term in any analysis.
> >
> > - **Please review the appropriate use of citep and citet.**
> >
> > Thank you for the suggestion, it certainly helped to improve clarity and readability of the paper.

---

> > > ### Comment · Reviewer_HR3f · 2024-07-05
> > >
> > > Thanks to the authors for their response, which has largely addressed my concerns .

---

### Review · Reviewer_vp5T · 2024-06-25

**Summary Of Contributions:**

This paper explores the gradient splitting interpretation of Temporal Difference (TD) learning and applies the Stochastic Variance Reduced Gradient (SVRG) method to achieve a geometric convergence rate. The authors present an algorithm that aligns with the convergence rate of SVRG in convex optimization by using a fixed learning rate of $1/8$. Their analysis spans three sampling regimes: finite sample, i.i.d., and Markovian settings. Empirical results are provided to support the claims.

**Audience:**

Yes

**Broader Impact Concerns:**

No Broader Impact Statement is needed for this work.

**Claims And Evidence:**

No

**Requested Changes:**

- Please address the weaknesses in detail for theoretical and empirical results
- Please provide intuition on how this new proof technique improves convergence in reducing the Bellman error
- Please comment on how this idea could be useful/intuitive in future research directions

**Strengths And Weaknesses:**

**Strength**

- The key idea is clearly described which makes it easy to follow.
- The idea allows shaving off the extra quadratic scaling on the condition number(which is not defined)/smallest eigenvalue
- The empirical evaluations show a clear benefit of using this new "method" in terms of convergence

**Weaknesses**

- Even though the manuscript appears to be mostly theoretical, the technical novelty seems low. The reviewer agrees that the gradient splitting idea allows easier analysis and improved results, however, the results of the paper follow a repetitive proof strategy borrowed from results in the literature. For example, Lemma 5.1 seems like the key tool that allows the analysis to go through and it mostly follows --as stated by the authors as well -- Johnson & Zhang (2013). The reviewer does not think this is a major issue as long as the proof technique provides a new insight into the core of the problem, i.e. learning the value function. However, the manuscript fails to highlight how this idea directly helps the convergence besides making the proof easier. Similarly, it is not clear whether this idea allows a novel algorithmic structure or development. The proposed Algorithm 1 (and its variants for different settings) seems trivial and lacks novelty. The authors should clarify these points and comment on possible future directions for how this idea can be used in RL.

- Even though the empirical results look impressive in terms of performance and reducing the batch size, it is not clear if the batch size is the only computational bottleneck in the performance. Each epoch of the algorithm, requires computing $\frac{1}{N} \sum_{s, s^{\prime} \in \mathcal{D}} g_{s, s^{\prime}}(\tilde{\theta}) $ for growing N. This will be the main bottleneck in the computation. This should be included in the empirical results to demonstrate the clear benefit of the presented algorithm. Also, it is not clear why VRTD is not compared in the empirical performance.

---

> ### Author Response · Authors · 2024-06-26
> **Answer to reviewer vp5T**
>
> We fundamentally disagree with the reviewer.
>
> - **The reviewer agrees that the gradient splitting idea allows easier analysis and improved results...**
>
> Indeed! The whole point of our paper is easier and more streamlined analysis and improved results.
>
> - **the results of the paper follow a repetitive proof strategy**
>
> Correct, the same high level idea works for many different settings. We view this as a positive outcome, consistent with a more streamlined analysis.
>
> - **For example, Lemma 5.1 seems like the key tool that allows the analysis to go through and it mostly follows --as stated by the authors as well -- Johnson \& Zhang (2013).**
>
> We were able to show how to make similar techniques used in the optimization literature (Johnson \& Zhang) work in the TD setting. Previous work attempted to invent new techniques and obtained worse results. Furthermore, Korda \& La (2015), Xu (2020) and Ma (2020) have technical issues with their analyses.
>
> - **The proposed Algorithm 1 (and its variants for different settings) seems trivial and lacks novelty.**
>
> Previous work proved guarantees for a more complex and intricate algorithm which keeps track of lots of extra variables. We view it as contribution the fact that we are able to prove guarantees for the simplest algorithm one can write.
>
> **In summary:** This paper is the first work, which analyzes both online sampling cases correctly (thanks to clear analysis) and improves the results significantly. We fully agree with the reviewer that  our paper provides "easier analysis and improved results" and we fundamentally disagree with the apparent view that the proofs need to be very complex to have value. If the results are indeed better, then this is the key contribution and getting this with a simpler and more streamlined analysis is an added bonus.
>
> Please refer to the TMLR acceptance criteria (\url{https://www.jmlr.org/tmlr/acceptance-criteria.html}) specifically:
>
> *Crucially, it should not be used as a reason to reject work that isn't considered “significant” or “impactful” because it isn't achieving a new state-of-the-art on some benchmark. Nor should it form the basis for rejecting work on a method considered not “novel enough”, as novelty of the studied method is not a necessary criteria for acceptance. We explicitly avoid these terms (“significant”, “impactful”, “novel”), and focus instead on the notion of “interest”. If the authors make it clear that there is something to be learned by some researchers in their area from their work, then the criterion of interest is considered satisfied. TMLR instead relies on certifications (such as “Featured” and “Outstanding”) to provide annotations on submissions that pertain to (more speculative) assertions on significance or potential for impact.*
>
> - **Even though the empirical results look impressive in terms of performance and reducing the batch size, it is not clear if the batch size is the only computational bottleneck in the performance. Each epoch of the algorithm, requires computing $\frac{1}{N} \sum_{s, s^{\prime} \in \mathcal{D}} g_{s, s^{\prime}}(\tilde{\theta}) $ for growing N. This will be the main bottleneck in the computation. This should be included in the empirical results to demonstrate the clear benefit of the presented algorithm. Also, it is not clear why VRTD is not compared in the empirical performance.**
>
> The reviewer confuses two different cases: dataset sampling and online sampling. The expression provided by the reviewer ($\frac{1}{N} \sum_{s, s^{\prime} \in \mathcal{D}} g_{s, s^{\prime}}(\tilde{\theta})$) refers to the dataset case, where the dataset size $N$ remains fixed and geometric convergence is achievable.
>
> The reviewer is correct that in online sampling cases (both i.i.d. and Markovian sampling), the estimation batch size grows with each epoch. The impact of this growth on total complexity depends on specific factors such as the MDP, features, policy, and the desired accuracy $\epsilon$. Naturally, as $\epsilon$ approaches 0, the total complexity will be dominated by the term $\epsilon^{-1}$. Experimental results for online sampling cases (both i.i.d. and Markovian) are presented in the Appendix of the paper, and VRTD is among the methods included for comparison.

---

> > ### Comment · Reviewer_vp5T · 2024-07-16
> >
> > I thank the authors for the clarification on the empirical results. In light of being of "interest" to the researchers in the field, I agree that the analysis in this work satisfies the criteria of interest of TMLR.

---

### Decision · Action_Editor_tfuu · 2024-07-16

**Recommendation:** Accept as is

**Comment:**

The paper shows that a simple application gradient splitting and SVRG can result in a simplified TD method that achieves a geometric rate of convergence.  This builds off other work, but is a stand-alone analysis that whose claims are justified by theory and experiments, and has an audience amongst those interested in the convergence of TD, with an even broader audience due to the clarity of presentation acknowledged by the reviewers.  This work clearly meets the TMLR criteria, even if the reviewers have some qualms about its advance over precursor results.

**Audience:**

Yes.

**Claims And Evidence:**

Yes.